# A macroevolutionary common-garden experiment reveals differentially evolvable bone organization levels in slow arboreal mammals

Fabio Alfieri [1,2✉], Léo Botton-Divet[1], Jan Wölfer[1], John A. Nyakatura [1] & Eli Amson [3]

Eco-morphological convergence, i.e., similar phenotypes evolved in ecologically convergent taxa, naturally reproduces a common-garden experiment since it allows researchers to keep ecological factors constant, studying intrinsic evolutionary drivers. The latter may result in differential evolvability that, among individual anatomical parts, causes mosaic evolution. Reconstructing the evolutionary morphology of the humerus and femur of slow arboreal mammals, we addressed mosaicism at different bone anatomical spatial scales. We compared convergence strength, using it as indicator of evolvability, between bone external shape and inner structure, with the former expected to be less evolvable and less involved in convergent evolution, due to anatomical constraints. We identify several convergent inner structural traits, while external shape only loosely follows this trend, and we find confirmation for our assumption in measures of convergence magnitude. We suggest that future macroevolutionary reconstructions based on bone morphology should include structural traits to better detect ecological effects on vertebrate diversification.

[1] Comparative Zoology, Institute for Biology, Humboldt-Universität zu Berlin, Unter den Linden 6, 10117 Berlin, Germany. [2] Museum Für Naturkunde, Leibniz-Institut für Evolutions- und Biodiversitätsforschung, Invalidenstraße 43, 10115 Berlin, Germany. [3] Paleontology Department, Staatliches Museum für Naturkunde, Rosenstein 1–3, 70191 Stuttgart, Germany. ✉email: fabio_alfieri@yahoo.it

The long-standing issue of understanding how phenotypic diversity has evolved[1–3] involves disentangling extrinsic (i.e., ecological) and intrinsic (e.g., phylogenetic, genetic, developmental) effects. The latter determines the evolvability. This biological emerging property is the propensity to evolve in response to opportunity[4] and has become a major topic in evolutionary studies[4,5]. Studying evolutionary patterns associated with similar ecological conditions (i.e., opportunity), we are able to factor out extrinsic drivers[4]. The widespread eco-morphological convergence, i.e., independently evolved morphological similarity associated with convergent ecologies, reproduces this experimental design and was compared to a common-garden experiment by Jablonski[4]. Accordingly, differences in morphological convergence strength, often arising from the study of ecologically convergent lineages on phylomorphospaces, potentially reveal gradients of evolvability.

Given the potential of differences in convergence strength to highlight differences in evolvability, a fundamental role is played by convergence strength quantification, for which disparate techniques have been developed[6–9]. Their diversity reflects the different theoretical frameworks on which they are based since, surprisingly, no unanimous agreement on the definition of convergence, and accordingly the criteria to be used in order to detect the process, has still been achieved[6,10,11]. Our approach emphasises the necessary condition that the independently acquired similar morphological traits set apart ecologically convergent taxa from their ecologically distinct close relatives. As detailed in the Discussion, we prioritise these aspects over those of other widespread approaches in which convergence is instead detected only if taxa geometrically converge on phylomorphospaces[10].

Analyses of eco-morphological convergence frequently address bone morphology, investigating few skeletal traits[12,13]. This approach informs on evolvability variation among lineages but not on a smaller scale, i.e., among skeletal parts of individuals. It is limiting since differential evolvability of individual parts, i.e., mosaic evolution, is increasingly recognised as a major driver of phenotypic diversification[3,14,15]. Addressing this phenomenon and its underlying mechanisms may be crucial to explain how intrinsic effects drive the distribution of phenotypic diversity. Mosaicism has been detected among regions of the entire skeleton[3] or traits of the same skeletal element[14], often identifying heterogeneous convergence patterns[16,17]. Yet, a dimension of this process remains unexplored: does evolvability vary across different anatomical levels of bone organisation? Since the evolution of anatomical features is affected by different morpho-functional constraints at different spatial scales[18], it is conceivable that mosaicism is represented across this axis of bone morphological variation.

At different spatial scales, bone anatomy reveals distinct characteristics[19]. External gross morphology ('shape', hereafter) constitutes a first level. In recent decades, technological advances granted researchers increasingly straightforward access to a second, finer level: the structural distribution of osseous tissue within skeletal elements[20] ('structure', hereafter). Evolutionary patterns have been reconstructed based on these two levels separately[12,21,22], but no macroevolutionary studies have attempted to compare evolvability pattern of bone shape with that of bone structure.

Both shape and structure exhibit numerous traits associated with ecological adaptations[23–29], e.g., the short and robust limb bone shape of fossorial digging species[30,31] or the heterogeneously oriented trabecular struts within long bones of climbing species[28,29]. When one compares the two levels, though, shape is hypothesised to reflect ecology less directly than structure[32]. Crucially, bone shape determines how bones contact surrounding tissues, e.g., ligaments, tendons and muscles, and to maintain overall body proportions. These complex interactions being crucial for organismal physiology[33,34], large shape modifications would involve an extensive and potentially deleterious reorganisation of many interacting elements, a scenario that is expectedly infrequent (although not impossible[35]). On the other hand, structure is more prone than shape to adapt to species' lifestyles, e.g., to locomotor mechanical stresses[32,36,37]. In this framework, it is expected that shape modifications are less likely selected than structural changes during ecologically driven phenotypic evolution. In other words, shape is expected to be less evolvable than structure.

Noticeably, bone tissue, at both shape and structure levels, exhibits high ontogenetic plasticity, i.e., responds to environmental inputs during individuals' lifetime[32,37,38] and, along this temporal scale, structure has been hypothesised to adapt more than shape[32]. To date, it is challenging to understand whether specific ecologically driven bone features derive from ontogenetic or evolutionary acquisitions, especially when studying numerous species and considering that ontogenetic patterns themselves evolve[39]. Yet, we can hypothesise that an anatomical level showing higher ontogenetic plasticity, e.g., bone structure, may result in wider variation in individual populations, on which selection may positively act over evolutionary time scales, thus in turn affecting evolvability.

A small number of previous studies, all restricted taxonomically (i.e., xenarthrans[28], mustelids[25,40], squirrels[41], squirrel-related rodents[17,42–44]) compared the ecological signal borne by bone shape and structure. Establishing the ecological signal of two bone anatomical levels might provide clues about their potential to evolve through adaptive phenotypic convergence and, in turn, about their evolvability (see above). When these studies were undertaken qualitatively, they recovered shape as more ecologically driven than structure[17,25,40,42–44]. However, besides the non-quantitative approach, these assessments were based on different samples from different studies, designed without the intention to compare shape and structure evolution. A different sample size was noticed as a possible explanation for the apparently stronger ecological signal of shape compared to structure[25,40]. Indeed, when the ecological signal of shape and structure was instead derived from quantitative measures on a homogeneous sample, structural traits clearly arose as more ecologically driven than shape ones[28]. Besides supporting our expectation of shape being less evolvable than structure, this motivated our study, which adopts the same approach but in a macroevolutionary context.

Leveraging the potential of eco-morphological convergence to epitomise a macroevolutionary common-garden experiment[4], we compared shape and structure convergence strength in ecologically convergent lineages. We focused on extinct and extant mammals who independently evolved slow arboreality. These taxa spend most of their life in trees and are characterised by extremely cautious arboreal climbing, low metabolic rate and an activity budget dominated by rest/quiescence[27,28]. To quantify morphological convergence, we sampled slow arboreal as well as closely related, ecologically distinct ('non-slow arboreal' hereafter) mammals. We focused on two limb bones, directly interacting biomechanically with the environment and thus suitable to detect how the ecological signal is reflected by morphology[33,45], i.e., the humerus and femur. These two skeletal elements have been shown to reveal ecological adaptations in shape and structure[28,29,43,46–48].

In accordance with our expectations, the common-garden experiment here addressed reveals the stronger evolvability of bone structure compared to shape. Indeed, humeral and femoral structural traits discriminate slow arboreal mammals from their

non-slow arboreal close relatives. Moreover, structural features reflect stronger convergence and explain major convergence patterns of slow arboreal mammals. These findings may contribute to redirect vertebrate evolution reconstructions considering that, thus far, they mostly relied on bone external shape information.

## Results

To conduct the macroevolutionary common-garden experiment, we first needed to quantify humeral and femoral shape and structural features (Fig. 1) of ecologically convergent slow arboreal mammals and their close relatives (Fig. 2, Supplementary Fig. 1, references in Supplementary Table 1). To do so, we partitioned bone morphology into shape and structure data (Tables 1 and 2). To represent external morphology in its entirety, we collected humeral and femoral data for the level shape through a high-density three-dimensional geometric morphometric (3DGM) approach. Both humeral and femoral shape were represented by the first ten Principal Components (3DGM PC1-PC10) deriving from generalised Procrustes analysis and principal component analysis (PCA). As for structural data, we accounted for the distinct features shown by long bones, as the humerus and the femur, from one extremity to the other (epiphyses), along their central shaft (diaphysis). To encompass all this variability, we built the dataset for the level structure including traits from different structural sub-regions, i.e., epiphyses (proximal and distal) and diaphysis (mid-diaphysis and average diaphysis), quantified through trabecular architecture and cross-sectional properties, respectively (Fig. 1). As epiphyseal trabecular traits, we extracted and analysed the degree of anisotropy (DA), trabecular thickness (Tb. Th), connectivity density (Conn.D), bone volume fraction (BV/TV), bone trabecular surface (BS/TV) and average branch length (Av. Br. Len). As diaphyseal cross-sectional properties, we extracted and analysed the global compactness (Cg), second moments of the area around the minor and the major axis (Imax and Imin), cross-sectional area (CSA) and cross-sectional shape (CSS). Once we extracted quantitative information on the humerus and femur of slow arboreal mammals, we aimed to detect distinct convergence patterns followed by different bone anatomical levels, the key aspect of the natural experiment here studied (see following 'Results' sections).

**Traits discriminating slow arboreal from non-slow arboreal mammals**. We expected that, as a sine qua non-condition, convergent traits in slow arboreal mammals should discriminate them from non-slow arboreal mammals. Hence, traits deserving further analyses of convergence are those that statistically differ between slow arboreal and non-slow arboreal mammals. These traits represented approximately one-third of all the studied traits (Tables 1 and 2, Supplementary Tables 2 and 3 and Supplementary Figs. 2 and 3). Most of the features (83%) within this subset concerned the structure level (Table 2). On the other hand, humeral and femoral shape discriminated slow arboreal taxa to a lesser extent (Supplementary Figs. 4 and 5). The low discriminant power of shape was particularly pronounced for the humerus, for which only one shape variable differed significantly between the two ecologies (Table 1). This was the 3DGM PC4, which represented just 4.85% of overall humeral shape variance (Supplementary Table 4). The femoral shape distinguished slow arboreal mammals through three variables (i.e., 3DGM PC2, PC6 and PC7; Table 1), cumulatively accounting for 16.5% of the entire femoral shape variance (Supplementary Table 5).

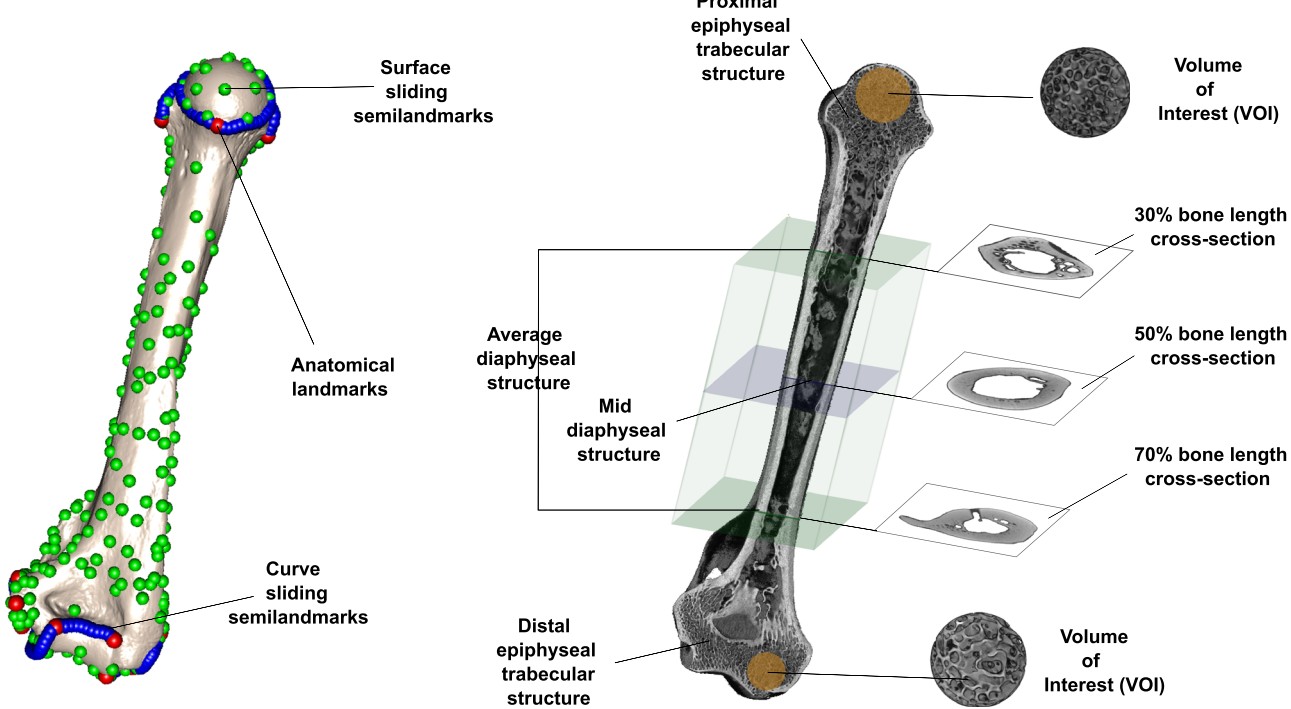

# Level 'shape'

Surface sliding semilandmarks

Anatomical landmarks

Curve sliding semilandmarks

# Level 'structure'

Proximal epiphyseal trabecular structure

Volume of Interest (VOI)

30% bone length cross-section

Average diaphyseal structure

Mid diaphyseal structure

50% bone length cross-section

70% bone length cross-section

Distal epiphyseal trabecular structure

Volume of Interest (VOI)

**Fig. 1 Partitioning of bone morphology into shape and structure data.** For the studied species, the shape was quantified through landmarks and 3D geometric morphometrics, while the structure was quantified at different structural sub-regions, through the extraction of volumes of interest (VOIs) of epiphyseal trabecular bone and cross-sections of diaphyseal bone. In the figure, data extraction is exemplified on the humerus of *Perodicticus potto* NMW 32674.

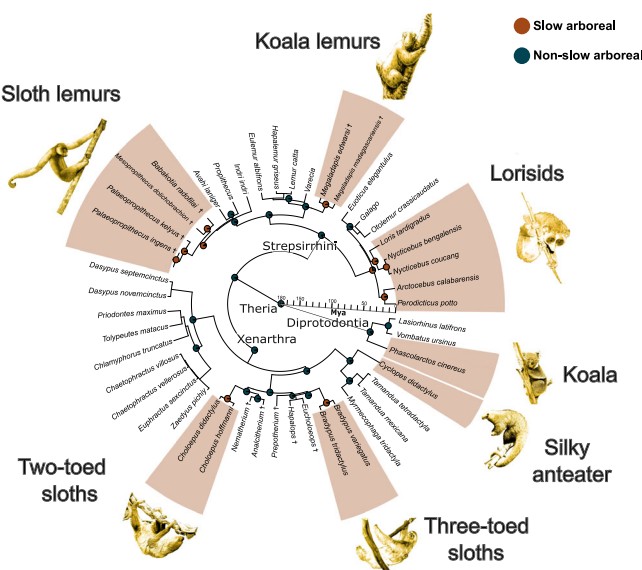

**Fig. 2 Convergent evolution of slow arboreality in mammals.** The seven independent acquisitions of the slow arboreal lifestyle (red sections), reconstructed with Stochastic Character Mapping, are shown on the time-calibrated tree of the taxa for which morphological data were obtained (see Supplementary Fig. 1 for an extended version). Timescale unit: million years ago (Mya). See Supplementary Note 4 for credits and copyright licenses for slow arboreal mammal images used in the figure.

**Table 1 Significant differences in external shape variables between slow and non-slow arboreal mammals.**

| Humerus | | Femur | |
| --- | --- | --- | --- |
| **Variable** | **Eco p** | **Variable** | **Eco p** |
| 3DGM PC1 | 0.071 | 3DGM PC1 | 0.831 |
| 3DGM PC2 | 0.458 | 3DGM PC2* | 0.013 |
| 3DGM PC3 | 0.675 | 3DGM PC3 | 0.069 |
| 3DGM PC4* | <0.001 | 3DGM PC4 | 0.764 |
| 3DGM PC5 | 0.583 | 3DGM PC5 | 0.512 |
| 3DGM PC6 | 0.408 | 3DGM PC6* | 0.004 |
| 3DGM PC7 | 0.706 | 3DGM PC7* | 0.045 |
| 3DGM PC8 | 0.860 | 3DGM PC8 | 0.086 |
| 3DGM PC9 | 0.097 | 3DGM PC9 | 0.672 |
| 3DGM PC10 | 0.488 | 3DGM PC10 | 0.081 |

The shape variables (i.e., first ten Principal Components, 3DGM PCs, resulting from generalised Procrustes analysis and PCA on landmarks coordinates) that differ between slow arboreal and non-slow arboreal mammals are highlighted with an asterisk. These traits were detected through PGLS and pANCOVA when the *p*-value related to the ecological category ('Eco p') was significant (<0.05). See Supplementary Tables 2 and 3 for a complete statistical summary and exact sample sizes for each test.

**Degrees of convergence reflected by slow arboreal mammal traits.** We grouped traits significantly discriminating slow arboreal mammals (Tables 1 and 2) for each skeletal element. Within the humeral and femoral datasets, a next subdivision concerned the anatomical level, i.e., shape traits vs. structural traits. To account for potential patterns of differential contribution across the heterogeneous structure of long bones, structural traits were further subdivided into datasets corresponding to the structural sub-regions (Fig. 1 and Table 2), all belonging to the structure level. To compare convergence strength among all these datasets, we quantified three indices, named C1–C3[8,10] (see 'Discussion' and 'Methods'). They overall inform on the extent to which a set of morphological traits is convergent in a group of taxa, focusing on the magnitude of phenotypic change that leads putatively convergent taxa to evolve similar morphologies. C1 was developed to quantify the proportion of the estimated maximum phenotypic distance that has been decreased by convergence. C2

**Table 2 Significant differences in inner structural variables between slow and non-slow arboreal mammals.**

| | Humerus | | Femur | |
| --- | --- | --- | --- | --- |
| | **Variable** | **Eco p** | **Variable** | **Eco p** |
| Mid-diaphyseal structure | log-Cg$_{50}$* | 0.016 | log-Cg$_{50}$* | 0.035 |
| | log-Imax$_{50}$ | 0.090 | log-Imax$_{50}$ | 0.132 |
| | log-Imin$_{50}$* | 0.040 | log-Imin$_{50}$* | 0.035 |
| | log-CSA$_{50}$ | 0.150 | log-CSA$_{50}$* | 0.029 |
| | log-CSS$_{50}$ | 0.660 | log-CSS$_{50}$ | 0.575 |
| Average diaphyseal structure | log-Cg$_{Aver}$* | 0.009 | log-Cg$_{Aver}$* | 0.018 |
| | log-Imax$_{Aver}$ | 0.090 | log-Imax$_{Aver}$ | 0.060 |
| | log-Imin$_{Aver}$* | 0.019 | log-Imin$_{Aver}$ | 0.067 |
| | log-CSA$_{Aver}$ | 0.078 | log-CSA$_{Aver}$* | 0.034 |
| | log-CSS$_{Aver}$ | 0.677 | log-CSS$_{Aver}$ | 0.259 |
| Proximal epiphyseal trabecular structure | DA$_{prox}$* | 0.007 | log-DA$_{prox}$* | 0.012 |
| | log-Tb.Th$_{prox}$ | 0.735 | log-Tb.Th$_{prox}$ | 0.745 |
| | log-Conn.D$_{prox}$* | <0.001 | log-Conn.D$_{prox}$ | 0.441 |
| | BV/TV$_{prox}$ | 0.059 | BV/TV$_{prox}$* | <0.001 |
| | log-BS/TV$_{prox}$ | 0.250 | BS/TV$_{prox}$* | <0.001 |
| | log-Av.Br.Len$_{prox}$* | 0.002 | log-Av.Br.Len$_{prox}$ | NA |
| Distal epiphyseal trabecular structure | DA$_{dist}$* | <0.001 | DA$_{lat.con}$* | 0.002 |
| | log-Tb.Th$_{dist}$ | 0.839 | log-Tb.Th$_{lat.con}$* | 0.002 |
| | log-Conn.D$_{dist}$ | 0.075 | log-Conn.D$_{lat.con}$ | 0.298 |
| | log-BV/TV$_{dist}$ | 0.539 | log-BV/TV$_{lat.con}$ | 0.355 |
| | log-BS/TV$_{dist}$ | 0.903 | log-BS/TV$_{lat.con}$ | 0.210 |
| | log-Av.Br.Len$_{dist}$* | <0.001 | Av.Br.Len$_{lat.con}$ | 0.096 |
| | | | DA$_{med.con}$* | 0.003 |
| | | | log-Tb.Th$_{med.con}$ | 0.541 |
| | | | log-Conn.D$_{med.con}$ | 0.926 |
| | | | log-BV/TV$_{med.con}$ | 0.104 |
| | | | log-BS/TV$_{med.con}$ | 0.354 |
| | | | log-Av.Br.Len$_{med.con}$ | 0.325 |

The asterisk highlights the structural traits that differ between slow and non-slow arboreal mammals. These traits were detected through PGLS and pANCOVA when the *p*-value related to the ecological category ('Eco p') was significant (<0.05). Variables are grouped by the structural sub-region from which they come from. See Supplementary Tables 2 and 3 for a complete statistical summary and exact sample sizes for each test. The diaphyseal structure was analysed through global compactness (Cg), second moments of area around the minor and the major axis (Imax and Imin), cross-sectional area (CSA) and cross-sectional shape (CSS), while the epiphyseal structure was analysed through degree of anisotropy (DA), trabecular thickness (Tb. Th), connectivity density (Conn.D), bone volume fraction (BV/TV), bone trabecular surface (BS/TV) and average branch length (Av. Br. Len).

estimates the absolute amount of phenotypic evolution, quantifying how much lineages converged in morphology and providing hints on the evolutionary response of given morphological traits to ecology. C3 measures how much of the morphological evolution is due to convergence. When the tests associated with C1–C3 significantly reject the null-hypothesis (*p*-value < 0.05), these indices' values are directly proportional to convergence magnitude, allowing a quantitative assessment of convergence strength[8] and, in this work, of evolvability.

We performed 99 convergence analyses, i.e., three indices (C1, C2 and C3), each one computed three times, for each of the datasets (collectively eleven datasets) (as detailed in 'Methods'). The analyses of the humeral structure datasets (i.e., considering all the sub-regions), yielded the highest proportion of significant C-indices: e.g., convergence analyses on all the diaphyseal datasets yielded 89% of significant C-indices, and the 78% of the C-indices computed on epiphyseal structural datasets (including both proximal and distal) were significant. Conversely, just 33% of the convergence analyses concerning the humeral shape yielded significant C-indices. The pattern of more frequently significant C-indices for structural datasets is also mirrored by the femoral data's distribution: 50% and 63% of significant C-indices yielded by the analyses of diaphyseal and epiphyseal datasets, respectively, against the 44% resulting from convergence analyses on the femoral shape datasets. Two-thirds of the C1 indices overall computed on both humeral and femoral structural datasets were significant, while across the humeral and femoral shape datasets, C1 was significant only in one case. Apart from this exception, C2 was the only significant index for the shape data. Hence, only C2 results were suitable to directly compare convergence magnitude values among shape and structural levels. C2 indices resulting

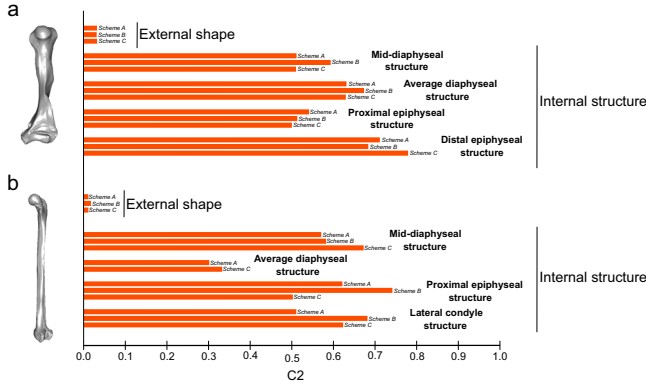

**Fig. 3 Shape *vs.* structure convergence strength in the humerus and the femur of slow arboreal mammals.** C2 values resulting from convergence analyses performed through 'convevol'[8,10], on **a** humeral and **b** femoral datasets. C2 is the only index for which results are shown in the figure, since it is the only for which shape and most of structural datasets yielded significant *p*-values (<0.05), hence returning informative values. Schemes A–C refer to convergence analyses repeated to account for potential biases (as detailed in 'Methods'). Results for the femoral medial condylar structural dataset and the Scheme B analysis of the femoral average diaphyseal structural level are not presented here because they did not yield significant *p*-values for C2 tests (Supplementary Table 6).

from shape data were close to zero in all the humeral/femoral convergence analyses (<0.035), while C2 indices ranging from 0.305 to 0.783 were associated with the structural levels (Fig. 3 and Supplementary Table 6).

**Convergence patterns in slow arboreal mammals.** We visualised convergence in slow arboreal mammals through phylomorphospaces. They are based on PCs (here named PC$_{pms}$), extracted from PCA on skeletal element datasets (Fig. 4). Slow arboreal mammals mostly occupy distinct regions from those of non-slow arboreal mammals, along PC$_{pms}$ of the humeral and femoral phylomorphospaces, although some overlap is present. *Bradypus* spp., *Choloepus didactylus* and 'Lorisidae' mostly occupy a region of the slow arboreal morphospace that is the farthest from the region of non-slow arboreal species. Subfossil lemurs mirror this pattern for femoral data. Most of the evolutionary trajectories of slow arboreal mammals point to the same region, reflecting the patterns of convergence (Fig. 4 and Supplementary Figs. 6 and 7).

In phylomorphospaces, angles and directions of the variable vectors reveal that slow arboreal mammal occupy of the same, distinct morphospace region mainly because of a subset of traits (Fig. 4b, d). They almost entirely concern structural aspects, i.e., highly interconnected (higher Conn.D$_{prox}$) and heterogeneously directed trabeculae (lower DA$_{med.con}$, DA$_{lat.con}$, DA$_{prox}$, although with a weaker contribution of the latter as shown by a shorter vector), low femoral trabecular surface (lower BS.TV$_{prox}$), and, in humeri and femora, more compact diaphyses (higher Cg). Three shape data traits seem to set apart slow arboreal mammals in the phylomorphospaces, i.e., femoral 3DGM PC2, humeral 3DGM PC4 and femoral 3DGM PC6 (the contribution for the latter two is minor).

## Discussion

In the last decades, convergence became apparent as a rampant and ubiquitous evolutionary process, in fact overturning the traditional view that saw it as marginal or occasional[49]. This renewed attention has been reflected by a plethora of studies addressing diverse aspects of convergence[8,50,51] including its contribution to elucidate broader issues in biology, e.g., how

evolvability varies in nature[4]. Eco-morphological convergence mirrors a macroevolutionary common-garden experiment[4] and enables us to use convergence patterns as indicators of evolvability to detect mosaic evolution.

Leveraging the potential of eco-morphological convergence to identify differences in evolvability, we highlighted the stronger evolvability of bone structure compared to shape in the humerus and femur of convergently evolved slow arboreal mammals. We found that structural traits prevalently set apart slow arboreal mammals (Tables 1 and 2) and are in, most cases, associated with stronger convergence (Fig. 3 and Supplementary Table 6). We identified an overall pattern of humeral and femoral convergence of some slow arboreal mammals on phylomorphospaces, as highlighted by taxa occupying the same sub-region in both the plots (see 'Discussion' below). Structural features (four humeral and five femoral traits, Fig. 4 and below) are those that mainly contribute to the overall humeral and femoral patterns of convergence in slow arboreal mammals. All results are in agreement with the expectation that convergence and evolvability in the humeral/femoral shape of slow arboreal mammals are weaker than in these bones' structure. The strong ecological signal in the convergent structural traits is consistent with the functional role associated with these traits in slow arboreal mammals. A low degree of trabecular alignment (DA) is a noticeable convergent trait. DA being related to the directional variability of biomechanical loadings, low DA is expected within skeletal elements experiencing highly variable mechanical environments[29], as those of slow arboreal climbers[52,53]. Higher diaphyseal structural compactness (Cg) of slow arboreal mammals may be explained by a weak selection against bone deposition in order to limit skeletal mass. In other words, no substantial selection for an optimal mass-saving phenotype[54] would be present in the humerus and the femur of slow arboreal mammals. While this selective pressure is likely essential in terrestrial and/or active vertebrates due to their lifestyles, slow arboreal mammals may have been freed from these constraints, convergently (by their peculiar ecological adaptations). The potential functional implications of highly interconnected trabeculae (Conn.D) and low trabecular surface (BS.TV) are less clear, but both traits have been found to reflect ecology[28,29].

A pattern involving the absence of convergence potentially results from two alternative scenarios: (i) the studied taxa show quite different morphological traits, e.g., divergence, (ii) the studied taxa share similar morphotypes that are not distinct, though, from those of their closely related taxa, i.e., phylogenetic conservatism. We posited that the overall weak convergence for the bone shape of slow arboreal mammals should be explained by the stronger anatomical constraints imposed on shape diversification. As a corollary, it should be consistent with a framework of phylogenetic conservatism. Humeral/femoral shape variability in the analysed taxa clearly supports this pattern, thus corroborating our hypothesis. Indeed, arboreal mammals, regardless of their 'slow' adaptations, are overall discriminated by humeral/femoral shape from non-arboreal mammals (PC1 in Supplementary Figs. 4 and 5). Importantly, for four out of the seven slow arboreal lineages studied here, the independent events of transition to this ecology would have occurred from arboreal, yet not slow, ancestors. 'Lorisidae', palaeopropithecids and *Megaladapis* possibly transitioned to slow arboreality from ancestors featuring active clinging and leaping, inferred as ancestral in strepsirrhine primates[55], and the koala probably evolved its distinctive lifestyle from a more generalist arboreal ancestor[56,57]. Thus, the fact that most of the humeral/femoral shape variation of these slow arboreal mammals does not evidently deviate from the other arboreal taxa's variation suggests conservative and highly intrinsically constrained bone shape evolution. Also, the femoral shape

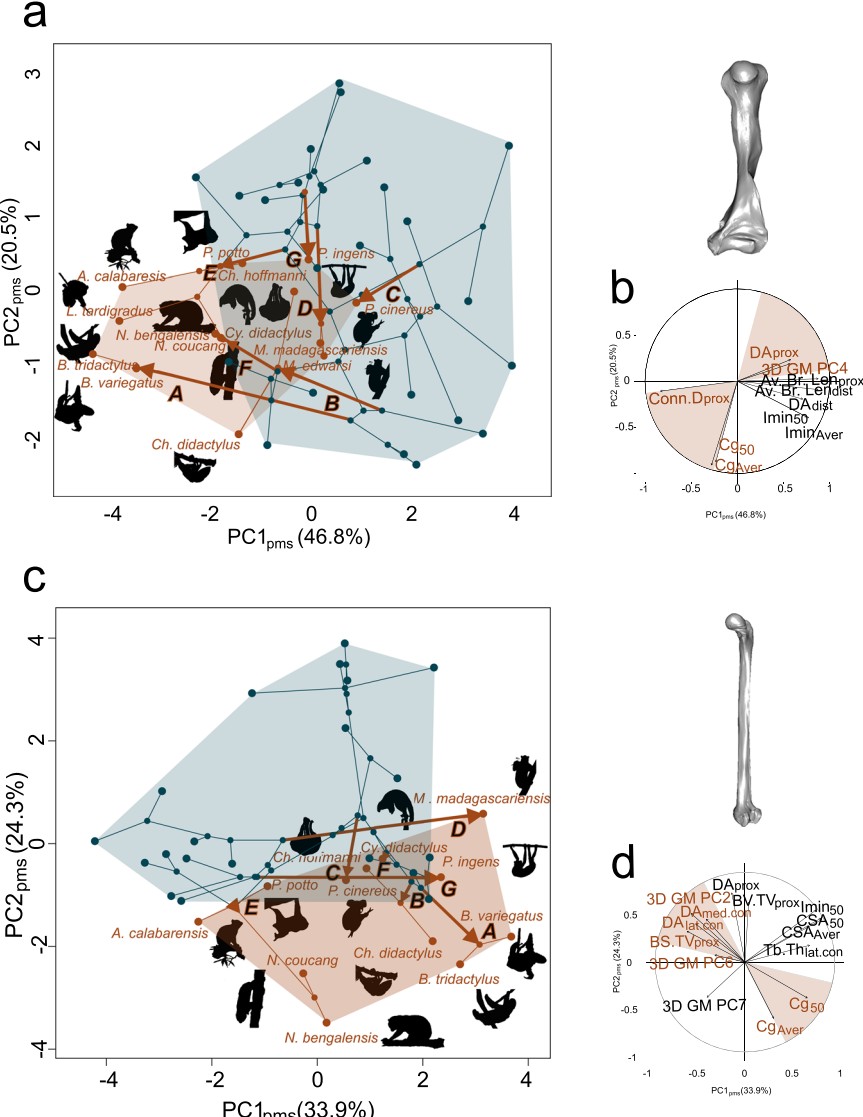

**Fig. 4 Overall patterns of humeral and femoral convergence in slow arboreal mammals.** In phylomorphospaces, taxa positions reflect morphology: closer taxa are more similar to each other. Hence, the relatively distinct position of slow arboreal mammals (light red) compared to non-slow arboreal mammals (blue) in phylomorphospaces (**a**, **c**, for humeral and femoral data, respectively) demonstrates that they tend to resemble each other. Moreover, phylomorphospaces allow to reconstruct trajectories of phenotypic evolution. In the figure, in each phylomorphospace the trajectories for the studied slow arboreal taxa (red arrows) are labelled with upper case letters: A—three-toed sloths, B—two-toed sloths, C—koala, D—koala lemurs, E—lorisids, F—silky anteater, G—sloth lemurs. As expected for convergent taxa, several convergent slow arboreal mammals occupy a distinct region of phylomorphospaces. In variable loadings plots (**b**, **d**, for humeral and femoral data respectively), traits mostly contributing to slow arboreal mammal convergence (orange) are identified evaluating their vectors' angles (informing on the trait-PC$_{pms}$ correlation strength), directions (telling if the trait-PC$_{pms}$ correlation is positive or negative) and lengths (reflecting how the trait contributes to taxa distribution on the plot). PC1$_{pms}$-PC3$_{pms}$ and PC2$_{pms}$-PC3$_{pms}$ biplots and related variable loadings plots are provided in Supplementary Figs. 6 and 7.

of the koala's sister taxon, i.e., the fully terrestrial and fossorial wombats (*Vombatus ursinus* and *Lasiorhinus latifrons*), lies within the morphospace occupied by arboreal taxa (Supplementary Fig. 5). It mirrors what has been found in a recent analysis of mammalian femoral shape, i.e., wombats occupy a morphospace region that is very close to that of arboreal taxa[58], and supports both the arboreal ancestry for the koala and the strong phylogenetic constraints posed to bone shape diversification. The latter is also supported by the silky anteater's humeral shape, which is distinct from that of other slow arboreal mammals and more similar to that of other anteaters (Supplementary Fig. 4), probably reflecting historical/phylogenetic constraints[28].

By addressing evolutionary patterns in slow arboreal mammal bone morphology, we also contributed to a better understanding

of this peculiar eco-morphological adaptation. Indeed, the multivariate phylomorphospaces convey how the different slow arboreal taxa contribute to the overall trend. *Bradypus* spp., *Choloepus* (especially *C. didactylus*) and 'Lorisidae' are the most distinctive slow arboreal mammals (Fig. 4 and Supplementary Figs. 6 and 7). Noticeably, these taxa share a suspensory locomotor/postural behaviour (observed to be sometimes used by 'Lorisidae'[59,60]) that was also inferred for palaeopropithecids[61]. Because *Palaeopropithecus* invades the slow arboreal region of the femoral phylomorphospace, departing from the one occupied by its ancestors, we infer that suspensory slow arboreal mammals are the main contributors to the overall convergence pattern. The suspensory lifestyle involves a distinct orientation of the body relative to the support, i.e., the centre of mass located below the

point of contact with the support, and a reversed gravitational loading regime, entailing a unique biomechanical environment[62,63]. These features make the suspensory lifestyle particularly specialised and the corresponding ecological niche substantially restricted. The functional demands of a suspensory lifestyle potentially imply a single adaptive optimum, that may cause a marginal role of phylogeny and strong eco-morphological convergence[64]. The other slow arboreal mammals, i.e., silky anteater, koala and koala lemurs, are possibly associated with a less specialised ecological niche, since they are characterised by more diverse locomotor behaviours. The silky anteater is a slow arboreal species characterised by vertical climbing but it occasionally sports a peculiar arboreal digging[65,66]. The locomotor ecology of the extinct koala lemurs has been reconstructed as slow arboreal vertical climbing[67], but the unavoidable biases involved in locomotor inference prevent us to depict a detailed range of locomotor habits. The slow arboreal locomotor style of the koala is also quite distinctive, because it is a robust animal, that uses claws, hands and feet in apposition of both sides of the support and the peculiar trunk-hugging behaviour; moreover, although non-frequent, some travels on the ground are reported[68]. The contribution of the silky anteater, the koala and the koala-lemur to the overall convergence pattern is probably due to the occasional convergence of a few traits of isolated species to suspensory slow arboreal species.

The possibility that shape, compared to structure, is less evolvable due to stronger anatomical constraints and phylogenetic conservatism may be a factor explaining that the modifications in structure are acquired earlier than those in shape when clades invade new ecological niches[69]. Hence, the stronger is the tendency of structure to evolve as a result of ecological specialisation, the stronger is its convergence magnitude, as we propose here. On a shorter timescale, another chronological perspective that may help to explain discrepant evolvability, and hence convergence, between shape and structure is ontogeny. Modifications of ontogenetic trajectories are the prevalent mechanism and the proximate cause for morphological diversification[2,70]. In the framework that we propose, ecologically convergent taxa would be characterised by somewhat decoupled ontogenetic patterns for bone shape and structure, with the former appearing to be more limited by anatomical constraints and the latter freer to adaptively evolve and thus also freer to converge. This model may be confirmed by evolutionary analyses of morphological change along ontogenetic stages and at different anatomical levels. This unexplored approach (ontogenetic studies of bone morphology thus far have exclusively focused on single anatomical levels of analysis[71,72]) may crucially contribute to describe multileveled eco-morphological convergence patterns. The decoupled developments of shape and structure would be achieved following specific patterns of phenotypic covariation, known to potentially have a major impact on adaptive diversification and evolution of complex morphologies[73,74]. Phenotypic covariation is often studied using the concepts of modularity and integration[75]. Modularity refers to the subdivision of phenotypic variation in quasi-independent blocks of traits named modules. Each module is built by a set of traits that are highly correlated to the point that they are almost autonomous relative to other traits belonging to other modules. Integration refers to the magnitude and pattern of covariance of traits within a single module[76,77]. Modularity and integration, potentially affecting evolution magnitude and morphological convergence[78,79], may explain the differential responses of shape and structure to convergence. In the bone system, shape and structure might represent two distinct modules, each constituting highly integrated traits, and their relative independence would be reflected by discrepant evolvability.

The strong tendency of structure to evolve in response to ecological opportunity has substantial implications for the reconstruction of vertebrate diversification. If this pattern can be generalised to a broader clade, it should orient future investigations to crucially include structural characteristics. Reconstructions of vertebrate morphological evolution based on bone anatomy have almost exclusively focused on outer shape[3,21,22,80]. Hence, they could have underestimated the role of ecological shifts. Great climate and geological changes, resulting in major ecological reorganisation, have possibly driven evolutionary rates and patterns[80]. A consequence of the underutilisation of structural traits in evolutionary inference is that the impact of these dramatic events may have been even more severe than previously estimated. In support of this assumption and considering that convergent evolution is diagnostic of ecologically driven diversification, the first macroevolutionary study of bone structure recently yielded a prominent role of convergence in the morphological evolution of mammals[12].

In this work, convergence magnitude was assessed through the C-indices ('convevol' R package[8]). Grossnickle et al. recently argued that C-indices, as well as all convergence measures, are potentially affected by some issues, i.e., scores are inflated if convergent taxa are outlying on the morphospace and convergence may be detected for lineages showing evolutionary trajectories that are not strictly convergent[10]. These scenarios do not fully align with their working definition of convergence, i.e., lineages that evolve to be more similar to each other compared to how similar their ancestors were[8,10,81,82]. To mitigate these effects, updated indices, i.e., Ct indices, have been developed[10]. Noticeably, these issues do not concern methodological limitations that would render C-indices outcomes meaningless (e.g., programming bugs), but rather turn out to relate to the theoretical framework of eco-morphological convergence analyses. Despite considerable efforts, an agreement on the criteria for identifying convergence has still to be reached[8,11,81,83]. Ct indices rely on a strict, geometry-based definition: focal taxa are convergent only if their evolutionary trajectories converge in the morphospace, i.e., the phenotypic distance between focal taxa is shorter than the distance between their ancestor[10]. However, we believe that in many comparative analyses, as this work, a more crucial role should be attributed to the focal taxa's position in the morphospace relative to the rest of the tree, i.e., the taxa considered to be unaffected by the adaptive regime under study. This can be achieved interpreting C-indices in combination with an examination of the morphospace. As detailed below and in Supplementary Fig. 8, it explains why we preferred the use of C-indices over Ct indices.

The core concept at the base of all definitions of convergence is independently evolved phenotypic similarity[8]. To claim that such similarity of convergent lineages is biologically meaningful, a necessary condition is that their phenotype is significantly different from that of their close relatives for whom a different adaptive regime is assumed (which in turn will affect the reconstructed ancestral values). In this framework, focal lineages that independently cluster within a distinct common region of the morphospace, potentially epitomise similar evolutionary responses to similar environmental pressures[84], i.e., the key tenet of convergent evolution. However, this scenario may not be considered convergence following the strict definition of Grossnickle et al. at the base of Ct indices[10]. For instance, in the simple case in which the distance between reconstructed ancestral trait values and the distance between observed values of focal lineages are measured on parallel lines, if the ancestral distance happens to be (even slightly) longer than the distance between observed values of focal lineages, the focal taxa are not considered convergent, under the geometry-based definition (see Supplementary Fig. 8).

We designed this entire study in order to focus on the morphospace positions of focal taxa relatively to the other taxa. Indeed, prior to convergence analyses, we selected traits that significantly discriminate slow arboreal from non-slow arboreal mammals. We recognise that trajectories on phylomorphospaces potentially provide important clues on evolutionary patterns, but we also believe that conclusions based on trajectories orientation alone, i.e., convergent taxa are more/less similar than were their ancestors, should be drawn with caution and in combination with additional evidence, e.g., data on extinct species. Indeed, trajectories orientation are crucially affected by ancestral phenotypes reconstructed assuming a Brownian motion model of evolution[8]. This model, based on random walk processes[85], oversimplifies the evolutionary patterns and unavoidably introduces biases, as Grossnickle et al. themselves have detected for Ct indices, as well as C-indices[10]. Since ancestral phenotypes are reconstructed under Brownian motion in the computation of C-indices, too[8], we advocate the use and interpretation of these indices in combination with an examination of the relative position of the focal and non-focal taxa, i.e., giving more importance to the actual, observed trait values.

Summarising, we here detected patterns of differential convergence among bone features at different spatial scales and interpreted this outcome as differences in evolvability among the studied anatomical levels. We achieved this aim by leveraging (i) the natural experiment provided by eco-morphological convergence, i.e., comparable to a common-garden experiment sensu Jablonski, (ii) the study system of the humerus and the femur and (iii) the case study of convergently evolved slow arboreal mammals. We highlight that the convergence magnitude of bone internal structure exceeds that of bone external shape, hence leading us to propose that the former is characterised by a stronger tendency to evolve. This pattern fully mirrors our expectation, i.e., a weaker convergence and evolvability of bone shape in comparison to internal structure, explained by stronger anatomical constraints and likely associated with phylogenetic conservatism. Our findings generalise previous observations made in more restricted phylogenetic contexts, which led us to propose that this pattern might be a common tendency broadly applicable to the phenotypic evolution of the skeletal system. Further investigations may contribute to elucidate the nature of the mechanisms behind the more conserved evolution of shape compared to structure, potentially following several perspectives, e.g., studies of ontogeny, modularity and integration.

## Methods

**Raw data collection**. We extended the dataset of ref. [28] following their bone specimen selection criteria. Namely, we pooled right and left skeletally mature humeri and femora (i.e., with fully fused epiphyses) from non-pathological and non-captive individuals. A few individuals (*Dasypus novemcinctus* FMNH 39307, *Indri indri* ZMB Mam-84278, *Nycticebus coucang* ZMB Mam-2718, *Propithecus* sp AMNH 170463) showed not complete fusion in some epiphyses (consequently not studied in their trabecular structure) but were included in the study since having the other epiphyses fully fused. Only for marsupials, adult stage was determined assessing the bone size, since not fully fused epiphyses are common in adult marsupials[86]. Overall, we analysed 109 humeri (including 9 isolated epiphyses) and 108 femora (including 8 isolated epiphyses) (Supplementary Data 1 and 2) collected from ten mammal collections in Austria, France, Germany, and the USA (Supplementary Note 1). We performed new taxonomical assignments for some specimens (Supplementary Note 2). Bones were scanned using micro-focus computed tomography (µCT) (Supplementary Note 3), generating image stacks (16-bit tifs)

with resolution adjusted to the size of the specimen (0.008–0.083 mm).

**Morphological data extraction**. In VG Studio Max 3.3 (Volume Graphics, Heidelberg, Germany), each humeral and femoral image stack was used to create a 3D mesh and was oriented with the *x*-, *y*- and *z*-axes aligned along the mediolateral, anteroposterior and proximodistal directions of the bones[28]. Humeral and femoral meshes were then post-processed in MeshLab[87] (simplification procedure, i.e., 'Ambient Occlusion' and 'Remove Vertices wrt Quality' tools, the latter with a threshold of 5%) and in Geomagic Wrap 2017 (3D Systems, Rock Hill, South Carolina, USA). On each 3D mesh, we captured the shape information. Bone shape can be measured through a traditional morphometric approach, based on linear distances, angles and functional indices[17,53], and geometric morphometrics (GM), based on landmark coordinates either in 2D (2D GM)[88,89] or in 3D (3DGM)[28,90]. Since our aim was an overall quantification of shape, we opted for a high-density 3DGM approach[90], i.e., locating anatomical landmarks + curve and surface sliding semi-landmarks (Fig. 1). This approach permits to maximise the captured morphological features, and for this reason was preferred over 2D GM. Moreover, through the approach it is possible to overcome issues related to the challenging recognition of key anatomical features, e.g., processes, in some of the studied taxa. This problem, known as ambiguous homology, may be exacerbated when the studied sample shows large variability[91], e.g., when several mammal clades are analysed, as in our case. Since the position and length of key anatomical characters are often involved in measures of distances and angles, ambiguous homology potentially prevents to adopt a traditional morphometric approach. Moreover, high-density 3DGM is particularly recommended when studying long bones, known for their scarcity of anatomical landmarks[92].

To employ this approach on mammals' humeri and femora, we largely followed the landmarking protocol of Alfieri et al.[28], summarised as following. We used their landmark sets consisting of 21 anatomical landmarks (exemplified in red in Fig. 1) + 195 curve semi-landmarks (exemplified in blue in Fig. 1) to represent humeral epiphyses and 22 anatomical landmarks + 254 curve semi-landmarks to represent femoral epiphyses[28]. These sets describing epiphyseal shape are the base for the procedure of sliding of surface semi-landmarks (exemplified in green in Fig. 1), allowing to capture additional shape information on the entire bone (see below). In MorphoDig 1.5.4[93], we positioned anatomical and curve semi-landmarks on humeral and femoral 3D meshes, so that two anatomical landmarks delimited a curve compounded by several curve semi-landmarks and that between any two adjacent curve semi-landmarks there was a homogeneous distance[94]. For landmarking, we mirrored left bones to have a sample of bones from the same side and we randomised the order of specimens. Damaged, deformed or incomplete bones were discarded (17 humeri and 16 femora, Supplementary Data 1 and 2). The humerus of *Eucholoeops* sp. FMNH P13280 and the femur of *Bradypus* sp. ZMB Mam-33806 (with the latter only used at this stage and not in the following analyses, as detailed in Supplementary Note 2) were selected as templates to drive semi-landmarks sliding on surfaces[91]. Using Blender[95], each of the models was manually inflated along the diaphysis and, additionally, only the humeral template locally smoothed to delete surface micro-cracks resulting from preservation of fossil specimens ('Inflate', 'Elastic Deform' and 'Scrape' tools). Then, the number of mesh triangles of the two templates were decimated in Geomagic allowing to visualise triangle vertices at approximately similar distances between the two templates. The processing steps performed in Blender and those in Geomagic

served to optimise the semi-automated process of sliding on the entire sample of bones and to drive the placing of surface semi-landmarks on the two templates, respectively. Surface semi-landmarks were placed on the two templates in MorphoDig. Semi-landmarks were not positioned in the olecranon fossa and in the region distal to the humeral head in posterior view (approximately until the neck level) on the humeral template and in the intercondylar fossa on the femoral template. These regions were excluded from landmarking because they are pronouncedly concave, a morphology that proved to produce uneven sliding with unwanted clustering of surface semi-landmarks. Finally, $n = 329$ and $n = 533$ surface semi-landmarks evenly covered the humeral and femoral template, respectively. In R 4.1.2[96]., we projected surface semi-landmarks from the templates on the other specimens ('placePatch' function, inflate = 10 for the humerus, inflate = 25 for the femur; Morpho package[97]) and we visually assessed the correct positioning on all the sample ('checkLM' function, Morpho package[97]). We iteratively slid curves and surface semi-landmarks employing the Procrustes consensus from the previous iteration and Thin Plate Spline bending energy minimisation between each specimen and the Procrustes consensus ('slider3d' function, Morpho package[97], setting; iterations = 20, stepsize = 0.5, recursive = TRUE, tol = 1e-8). Then, we performed generalised Procrustes analysis and principal components analysis (PCA) ('gpagen' and 'gm.prcomp' functions, geomorph package[98]) and we took PC1-PC10 (specified as '3DGM PCs') for both humeral and femoral data (explaining 89% and 85% of total variance, respectively, Supplementary Tables 4 and 5) to analyse humeral and femoral shape.

To quantify CSP we imported the oriented stacks in FIJI[99]. The diaphysis is defined as the region comprised between the 30% and 70% of the entire bone length from the proximal end (Fig. 1), since it is the longest diaphyseal portion for which epiphyses are completely excluded for all specimens. Non-directional diaphyseal CSP were measured exploiting the slice-by-slice approach of Amson[100], specifically employing the modified FIJI macro version provided by (and freely downloadable from) Alfieri et al.[28,101] (based on the 'Slice Geometry' tool of the FIJI BoneJ plugin[102]) and following Alfieri et al.'s workflow[28], as detailed below. After thresholding and purifying ('Optimise Threshold > Threshold-Only' and 'Purify' BoneJ routines) we computed global compactness (Cg, %), the second moments of area around the minor and the major axis (Imax and Imin, both having mm$^4$ as unit), the bone cross-sectional area (CSA, mm$^2$), and the cross-sectional shape (CSS, Imax/Imin, no unit). Diaphyseal single slices or intervals for which CSP were biased (due to bone integrity issues and/or fossil preservation), were excluded and replaced in R with values estimated from neighbouring non-biased slices (in 22 humeri and 25 femora). In bones subject to this correction, if the biased slices occupied an extremity of the diaphysis (preventing neighbours-based values reconstruction), they were manually restored in FIJI (in three humeri and four femora). Each CSP was averaged and extracted along the diaphysis (referred to as Parameter$_{Aver}$) and on the mid-diaphysis (50% of the bone length, reportedly the most informative level in limb long bones of mammals[103], here referred to as Parameter$_{50}$). Only Parameter$_{50}$ was extracted if the specimen showed dominant preservation issues along the diaphysis (in eight humeri and eight femora, with minor cracks at the 50% level manually repaired in five humeri and six femora). If these preponderant preservation issues involved the 50% level too, the specimen was discarded (two humeri and three femora). The correction procedure and specimens involved are further detailed in ref. [28] and Supplementary Data 1 and 2, respectively.

Trabecular parameters were computed on two (for the humerus) and three (for the femur) spherical volumes of interest

(VOIs), centred within the humeral head, capitulum (Fig. 1), femoral head, lateral and medial condyle, and extracted from oriented humeral and femoral stacks in FIJI. Each VOI represents the largest sphere that samples only trabecular bone. VOI extraction was performed through the FIJI macro provided by (and freely downloadable from) Alfieri et al.[28,101]. Femoral head VOIs were subsequently halved taking the lateral hemispherical VOI, due to the deep *fovea capitis* of *Myrmecophaga* and *Tamandua*, that would have caused empty regions in the medial side of the spherical ROI, biasing the trabecular computation. The diameter range for all the VOIs is 1–27 mm (Supplementary Data 1 and 2). VOIs showing damaged and/or dramatically biased trabecular structure (from seven humeral heads, 13 capitula, 13 femoral heads, 11 lateral condyles and 16 medial condyles, mostly from fossil specimens) and VOIs from unfused epiphyses of adult marsupials (six humeral heads, one capitulum, three femoral heads, one lateral condyle and one medial condyle) were discarded. Other VOIs were only moderately filled with visually distinguishable non-bone material. Those specimens (eight humeral heads, six capitula, six femoral heads, four lateral condyles and one medial condyle) were included after having been manually thresholded ('cleaned') in FIJI[28]. Supplementary Data 1 and 2 report the discarded and 'cleaned' specimens. In FIJI, we automatically thresholded all the other VOIs, we purified the entire VOIs sample and we extracted seven trabecular parameters through the respective BoneJ routines: degree of anisotropy (DA, no unit), trabecular thickness (Tb.Th., mm), connectivity (Conn., no unit, computed only to estimate the number of trabeculae, see below), connectivity density (Conn.D., i.e., Conn/Total Volume (TV), mm$^{-3}$), bone volume to total volume (BV/TV, no unit), bone surface to total volume (BS/TV, mm$^{-1}$) and average branch length (Av.Br.Len., mm; measured only after the skeletonization of the stack performed through the 'Skeletonise 3D' routine). Results for parameters that rely on the VOI total volume (i.e., Conn.D., BV/TV and BS/TV) were subsequently corrected considering a spherical volume, since BoneJ considers by default a cubic VOI. We refer to parameters from proximal epiphyses as Parameter$_{prox}$, to those from distal humeri as Parameter$_{dist}$ and to those from lateral and medial condyles as Parameter$_{lat.con}$ and Parameter$_{med.con}$, respectively. The relative resolution (Tb.Th/scan resolution[104]) of all VOIs ranges from 3.72 to 19 (average relative resolution of 7.84) (Supplementary Data 1 and 2), thus they can be considered reliable for trabecular analysis according to recommended values of ref. [104] and ref. [105]. The humeral head VOI of *Palaeopropithecus ingens* DPC UA5474 (3.96) and the femoral head VOI of *Prepotherium* sp. YPM-PU-15345 (3.72) have values slightly lower than those recommended but resolution and contrast were visually assessed and both VOIs validated. VOIs including less than 50 trabeculae (with Conn approximately representing the number of trabeculae) were discarded, following the suggestion of Mielke et al.[44]. Femoral Av.Br.Len$_{prox}$ was not further analysed since showing wide fluctuations and already discarded by Alfieri et al.[28] after an assessment of the repeatability of the entire data extraction protocol.

**Time-calibrated phylogeny and ancestral lifestyle reconstruction.** In order to infer evolutionary patterns, we built a time-tree of the sampled taxa. It is based on the Maximum Clade Credibility (MCC) DNA-only node-dated phylogeny of 4098 mammal species from the posterior distribution generated by Upham et al.[106]. The phylogeny of Upham et al. allowed us to avoid polytomies[106]. In Mesquite[107], we adapted Upham et al.' s MCC[106] time-tree to our dataset. First, we pruned all clades not studied in this analysis. Then, we adjusted the phylogeny in order to

accommodate our taxonomic assignments (see Supplementary Note 2): i.e., *Galago*, *Varecia* and *Propithecus* species were collapsed in three tips, one for each genus. Finally, we added taxa not represented in the phylogeny, i.e., Santacrucian sloth genera (*Hapalops*, *Eucholoeops*, *Nematherium*, *Analcitherium*, *Prepotherium*), *Dasypus septemcinctus*, *Tamandua mexicana*, *Mesopropithecus dolichobrachion*, *Babakotia radofilai*, *Palaeopropithecus maximus*, *Palaeopropithecus kelyus*, *Megaladapis madagascariensis*. Extinct sloth phylogenetic positions and times of divergence were taken from the phylogeny used by Alfieri et al.[27] (that, in turn, included extinct sloths combining other information, from Bargo et al.[108], Varela et al.[109] and Delsuc et al.[110]). *Dasypus septemcinctus* and *Tamandua mexicana* were added to the time-tree based on their most recent phylogenetic reconstructions[111,112]. *M. dolichobrachion*, *B. radofilai* and *P. maximus* were included following Herrera and Dávalos[113]. It was not possible to transform Upham et al.'s MCC tree[106] concerning the phylogenetic position and divergence time of Palaeopropithecidae only. Indeed, it would have resulted in inconsistency with Upham et al.[106] about the relationships with 'Indriidae'. Hence, Herrera and Dávalos' data were used to constrain topology and divergence times for the whole ('Indriidae'+Palaeopropithecidae) clade, involving the paraphyly of 'Indriidae'[113]. Since not represented in both Upham et al.[106] and in Herrera and Dávalos[113] studies, *Palaeopropithecus kelyus* was included in the phylogeny following data from Baab et al.[114], to our knowledge the only work that reconstructed phylogenetic position and divergence time for *P. kelyus*. Concerning the species of *Palaeopropithecus*, Baab et al.'s phylogeny only included *P. kelyus* and *P. ingens* (but not *P. maximus*), with their divergence time being 6.25 Mya[114]. The fact that this age is older than the *P. ingens*-*P. maximus* divergence time inferred by Herrera and Dávalos[106] constrains the topology of the *Palaeopropithecus* species here studied to *P. kelyus* being the sister taxon of the clade (*P. maximus* + *P. ingens*). *M. magadascariensis* was added as sister taxon of *M. edwarsi* with the divergence time taken from ref. [114]. Importantly, the phylogenetic position of *Megaladapis* according to ref. [106], thus that we relied on, mirrors what was recently found analysing the nuclear genome ancient DNA of *M. edwarsi*[115]. It is also noteworthy that Upham et al.'s tree implies that 'Lorisidae' is paraphyletic[106] (Fig. 2) (see also refs. [116] and [117]).

On the time-calibrated phylogeny, through Stochastic Character Mapping (SCM), we reconstructed the seven events of independent acquisition of a slow arboreal lifestyle. Following ref. [27], SCM was run on the studied taxa + twelve other species, to include a wider ecological diversity and reduce bias in the reconstruction (Supplementary Fig. 1), assigning to each tip either the 'slow arboreal' or the 'non-slow arboreal' state (Supplementary Table 1). We used the 'make.simmap' function (1000 simulations, 'phytools' R package[118,119]) with the 'equal rate' (ER) model. Alfieri et al.[27] compared the likelihood of the ER model and models in which reversion from slow arboreality is less probable than acquisition, a hypothesis based on the highly specialised adaptations involved in this ecology. Since the ER model was recovered as more likely[27], we chose it for SCM. The expected independent acquisitions of slow arboreality were confirmed as highly probable through SCM, justifying our approach. Slow arboreality was reconstructed as the ancestral state for 'Lorisidae' with galagids subsequently acquiring another ecology, as found by Alfieri et al.[27] (Fig. 2).

**Identifying traits discriminating slow arboreal mammals**. All the statistical tests used in this work are two-sided. In order to identify a subset of traits that are best candidates to be tested for convergence strength, we first selected the traits that significantly discriminated slow arboreal from non-slow arboreal mammals.

Moreover, it allowed us to decrease computational power as well as noise in multivariate analyses of adaptive convergence. Hence, we ran a series of univariate Phylogenetic Generalised Least Square (PGLS) regressions and phylogenetic ANCOVAs (pANCOVAs). Lifestyle was coded as a binary variable (i.e., 'slow arboreal' or 'non-slow arboreal') and a body size proxy was used as a covariate. As body size proxy, we took the natural log-transformed centroid size[120] from the configurations of anatomical landmarks + semi-landmarks resulting from generalised Procrustes analysis. For specimens discarded from 3DGM analysis we predicted the log-transformed centroid size exploiting its strong covariation with epiphyseal metric measurements and following Alfieri et al.'s[28] approach, that is detailed below. On each of the humeral and femoral epiphyseal regions from which we extracted VOIs of trabecular bone (see above) we measured three lengths, i.e., proximodistal, mediolateral and anteroposterior (Supplementary Data 1 and 2). The lengths of each epiphysis were averaged obtaining a single value for epiphysis, hence having two values (one proximal and one distal) for each complete humerus and two values (one proximal and one distal) for each complete femur. Through linear models we predicted the values for missing epiphyses in cases of incomplete specimens, entering the value of the other epiphysis in the respective model. Then, for each specimen we averaged the proximal and distal epiphyseal values obtaining one average epiphyseal value for specimen that, thus, was available for each bone, either complete or fragmentary. Finally, this average epiphyseal value was used to run a linear regression against the natural log-transformed centroid size, used to predict the latter for specimens discarded from 3DGM analysis. All the prediction procedures were performed with the 'predict' function, ('stats' R package[96]). Two specimens not analysed with 3DGM represent isolated epiphyses of the same bone of the same individual: i.e., proximal and distal femoral epiphyses of *Megaladapis madagascariensis* MNHN MAD-1564. Since the log-transformed centroid size is intended to represent a proxy for the bone size (and, ultimately, body size), the estimated log-transformed centroid sizes for these two specimens (very close but slightly different) were averaged and the same body size proxy was used for both specimens, consistently with their belonging to the same bone.

On each morphological trait, we separated the effects of lifestyle and body size (taken into account through its proxy, as detailed above), since they are not correlated (pANOVAs; $p$-value$_{hum}$ = 0.44, $p$-value$_{fem}$ = 0.26). Morphological traits were natural log-transformed if deemed necessary (Tables 1 and 2). Specifically, each PGLSs was run twice, using variable raw values and log-transformed values. PC scores (also represented by negative values) were all made positive adding a constant value (i.e., minimum variable value * 1.0001) to raw results, prior to log-transformation. This twofold PGLS allowed us to understand if log-transformation was necessary for each trait. It was done visually assessing the distribution of residuals from the two regressions[121] and preferring the condition resulting in residuals distribution closer to normality. In order to minimise transformation of raw data, in cases of similarly normal distribution of residuals between the two PGLSs for each trait, not log-transformed values were preferred. For each taxon, the average value for each trait as well as the average value of the body mass proxy were calculated. PGLS regressions were run using the 'gls' function ('nlme' R package[122]) while estimating Pagel's lambda (λ) ('corPagel' function, 'ape' R package[123]). Maximum likelihood was used by default and restricted maximum likelihood in case no model convergence was reached. The likelihood estimation method and λ values for each trait and complete results for each PGLS regression (i.e., parameter estimate, standard error, t-value and p value) are shown in Supplementary Tables 2 and 3. Mean taxa results for traits significantly setting

apart slow arboreal mammals are shown with boxplots (Supplementary Figs. 2 and 3). For variables found as significantly correlated with body mass (i.e., pANCOVA p-value for the covariate 'log-transformed centroid size' <0.05, Supplementary Tables 2 and 3), boxplots were built using size-corrected values. We size-corrected values using the residuals from a linear regression against the respective body mass proxy. Size correction was not done for shape variables i.e., PC scores, since it would have been in part redundant with Procrustes superimposition. We extracted the morphology represented by PC scores warping meshes to maximum and minimum PC score values ('warpRefMesh' function, 'geomorph' R package[98]). It was done for 3DGM PC1-PC2, to visualise most of the humeral and femoral shape variability (Supplementary Figs. 4 and 5).

**Convergence analyses.** We separated the log-transformed and/or size-corrected traits significantly setting apart slow arboreal from non-slow arboreal mammals (Tables 1 and 2 and Supplementary Tables 2 and 3) in humeral and femoral traits. Then, within the set of traits of each bone we grouped traits by anatomical level, i.e., 'shape' vs. 'structure'. Structural traits were further subdivided following their structural sub-region (Fig. 1 and Table 2), to account for inner morphological variability of long bones. Moreover, only for the femur, distal epiphyseal traits were pooled by the condyle to which they belong, i.e., medial vs. lateral. Overall, it resulted in five humeral (H1. 'shape'; H2. mid-diaphyseal; H3. average diaphyseal, H4. proximal epiphyseal, H5. distal epiphyseal; with H2-H5 representing the 'structure') and six femoral datasets (F1. 'shape'; F2. mid-diaphyseal; F3. average diaphyseal, F4. proximal epiphyseal, F5. lateral condyle epiphyseal; F6. medial condyle epiphyseal; with F2-F6 representing the 'structure') (Supplementary Table 6). Quantifying convergence strength in each of these datasets allows us to compare how strongly converge slow arboreal mammals in their humeral/femoral shape (datasets H1 and F1) vs. their humeral/femoral structure (datasets H2-H5 and F2-F6). For all the analyses, we employed the approach at the base of the C-indices of the 'convevol' R package[8,10], quantifying convergence strength for a set of traits in a group of a priori defined convergent taxa and testing whether this convergence strength differs significantly from simulated random evolution (Brownian motion as a null model). They are based on reconstructed past distances and the observed phenotypic distances[8,10]. Within 'convevol', we preferred the computation of C-indices over the computation of the recently developed Ct indices since they differ in the definition of convergence to which they refer (see detailed treatment provided in ref. [10] and 'Discussion').

Before running convergence analyses, we standardised and centred variables with different units/scales ('scale' R function) in multivariate datasets. Then, we excluded the signal due to similarity among taxa within the same lineage of slow arboreal mammals (e.g., between *B. tridactylus* and *B. variegatus* within *Bradypus*), since it results more probably from homology, i.e., inherited traits, rather than convergence. To do so, we averaged values of multiple taxa, obtaining average single trait values for each slow arboreal lineage. It was not done only for the paraphyletic 'Lorisidae', as it is impossible to collapse 'Lorisidae' species in a single tip without including the non-slow arboreal galagids (Fig. 2). We used single values for the monophyletic clades 'Nycticebus + Loris' and 'Arctocebus + Perodicticus', minimising the effects of intra-taxon similarity, yet not excluding them (see below). For each convergence analysis, the phylogeny was modified accordingly.

We used the approach of the 'convevol' functions 'calcConv' and 'convSig' to quantify C-indices and associated p values (based on 1000 simulations) on the five humeral and six femoral

datasets. Some datasets (e.g., H1. humeral shape) were univariate, since only one trait from that level or structural sub-region significantly discriminated slow arboreal from non-slow arboreal mammals (Tables 1–2, see above). In these cases, we adapted the 'convevol' functions to univariate analyses ('calcConv1D' and 'convSig1D'), as detailed in the R code (Supplementary Data 4). Setting as putatively convergent *Bradypus, Choloepus, Cy. didactylus, P. cinereus, Megaladapis*, Palaeopropithecidae, 'Nycticebus + Loris' and 'Arctocebus + Perodicticus' we ran a first set of analyses ('Scheme A'). To exclude potential biases related to 'Lorisidae' paraphyly and homology (see above), we repeated each analysis, first excluding 'Arctocebus + Perodicticus' ('Scheme B') then 'Nycticebus + Loris' ('Scheme C') (Supplementary Table 6). When C-indices showed significance for the shape and most of the structural levels, we visualised their C-values through bar charts (Fig. 3).

To visualise overall patterns of humeral/femoral phenotypic convergence, we built one dataset per skeletal element, joining the previously split sub-datasets (i.e., H1-H5 compounded the humeral dataset, F1-F6 compounded the femoral dataset). They represent the features of the bone phenotype (i.e., from different levels) that significantly set apart slow arboreal mammals. From Principal Component Analyses on both the skeletal element datasets, we extracted the first three Principal Components (PC), accounting for the 78.5% and 69.4% of the humeral and femoral variance, respectively (Supplementary Tables 7 and 8). They were used to build 2D (with pairs of PCs) phylomorphospaces ('phylomorphospace' R function, 'phytools'[119]) (Fig. 4 and Supplementary Figs. 6 and 7). To avoid confusion with the PCs resulted from shape analysis (i.e., 3DGM PCs), the PCs used to build phylomorphospaces are named $PCs_{pms}$. The reconstructed trajectories of phenotypic evolution for slow arboreal mammals were highlighted on 2D plots. Phylomorphospaces allow to assess main convergence patterns in slow arboreal mammals (through their position on the plot and trajectories' direction) and the contributions of single shape and structural traits (assessing angles, directions and lengths of their vectors) (Fig. 4).

**Statistics and reproducibility.** All the statistical observations are values corresponding to biological taxa, e.g., species, with data represented by the average of specimens belonging to the same taxon. Since taxa are not independent, i.e., they are phylogenetically related, all the statistical analyses were performed with phylogenetic comparative methods, that account for phylogenetic relatedness. In this work, replicates are represented by biological taxa characterised by the same lifestyle ('slow arboreal' or 'non-slow arboreal'). In PGLS regressions and ANCOVAs, the sample size, i.e., number of replicates, is $n = 13$–17 for 'slow arboreal' species and $n = 23$–27 for 'non-slow arboreal' species (there is variation because some species could be not represented due to specimen preservation issues).

All the bones analysed in this work are identified with a catalogue number in the collection from which they can be sampled (Supplementary Data 1 and 2). Most of the μCT scans generated from the specimens are freely downloadable from MorphoSource (see Supplementary Data 1 and 2 for the related ARK ID code that identifies the respective media) while a minor part of the virtual data will be made available upon request. Details on all the methodological aspects of the work are provided in 'Methods', Supplementary Information and the R code, making the whole study fully reproducible.

**Reporting summary.** Further information on research design is available in the Nature Portfolio Reporting Summary linked to this article.

## Data availability

Raw data from and additional information on the studied bones (Supplementary Data 1 and 2), time-tree (Supplementary Data 3) and Supplementary Information (including additional figures, tables and notes, besides credits and copyright licenses for slow arboreal mammal images used in Fig. 2), are available on Figshare (https://doi.org/10.6084/m9.figshare.22061207.v11)[124]. Image stacks and 3D meshes can be downloaded from MorphoSource (https://www.morphosource.org/projects/000393379, ARK ID codes in Supplementary Data 1–2), excluding those deriving from SMNS, Stuttgart, ZFMK, Bonn, ZSM, Munich, (all in Germany) and DPC, Duke University (USA) that are made available upon reasonable request.

## Code availability

The R code (Supplementary Data 4), developed in R 4.1.2[96] and including all the steps needed to repeat the analysis, is downloadable from Figshare (https://doi.org/10.6084/m9.figshare.22061207.v11)[124].

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

## Acknowledgements
We thank three anonymous reviewers for their precious suggestions that greatly improved the manuscript. This work was made possible by curators and assistant curators who allowed visits to collections and access to bone specimens: Frieder Mayer, Christiane Funk and Anna Rosemann (ZMB, Museum für Naturkunde, Berlin, MfN), Eva Bärmann (ZFMK), Frank Zachos and Alexander Bibl (NMW), Stefan Merker (SMNS), Anneke van Heteren (ZSM), Guillaume Billet (MNHN), Neil Duncan (AMNH), Sara Ketelsen (AMNH), Vanessa Rhue (YPM-PU), Daniel Brinkman (YPM-PU), Adam Ferguson (FMNH), William Simpson (FMNH), Matt Borths (DPC), and Catherine Riddle (DPC). Furthermore, we would like to thank Kristin Mahlow and Martin Kirchner (MfN), Renaud Lebrun (MRI-ISEM, Montpellier), Justin Gladman (SMIF, Durham, NC, USA), April Isch Neander and Zhe-Xi Luo (University of Chicago, IL, USA) for allowing access to micro-CT scanners and providing precious help. We acknowledge the MRI platform member of the national infrastructure France-BioImaging supported by the French National Research Agency (ANR-10-INBS-04, «Investments for the future»), the labex CEMEB (ANR-10-LABX-0004) and NUMEV (ANR-10-LABX-0020). This work was performed in part at the Duke University Shared Materials Instrumentation Facility (SMIF), a member of the North Carolina Research Triangle Nanotechnology Network (RTNN), which is supported by the National Science Foundation (Grant ECCS-1542015) as part of the National Nanotechnology Coordinated Infrastructure (NNCI). The study was funded by Elsa-Neumann-Stipendium des Landes Berlin, the German Research Council (Deutsche Forschungsgemeinschaft; grant number AM 517/1-1) and the Kickstarter Program from RTNN (NC, USA).

## Author contributions
F.A. collected the sample, acquired, processed, and analysed µCT data, performed statistical analyses, and drafted the manuscript. L.B.D. and J.W. contributed to the external shape analysis, E.A. to the internal structure analyses. F.A., L.B.D., J.W., J.A.N. and E.A. designed the study, interpreted the data, and contributed to the writing and editing of the manuscript.

## Funding

## Competing interests
The authors declare no competing interests.
