## [Peer Review File · Communications Biology]

Reviewers' comments:

Reviewer #1 (Remarks to the Author):

Having a strong personal interest in suspensory locomotor structure and function, I was particularly interested in reviewing this manuscript. My overall impression is favorable. I also believe that this work will make a valuable contribution to the current literature on bone structure and function. It certainly provides new perspectives of what levels of bone provide the greatest opportunity to observed important traits that can be evaluated for convergence or divergence. As opposed focusing on numerous minor comments and suggested edits, I will focus my review on major comments for the authors to consider.

1. I am quite impressed with the level care and critical attention to details with data analysis. However, I must admit that I am unfamiliar with several of the R 'toolkits' that the authors employed. Nonetheless, I am quite familiar with previous work of two of the authors, and I trust that the analysis was conducted correctly and the statistical methods were also applied correctly. I think that the tables and figures are useful for condensing the data. Several of them are indeed, well done. If I am to make one criticism here it would be that the size of the panels in Fig. 4 are a bit too small. The 3D-PCA phylomorphospace plots are accordingly difficult to interpret, and the 2D plots are quite garbled.

2. The authors did an excellent job citing referenced literature. In particular, terminology (e.g. mosaic evolution) is clearly defined and well-referenced. My main criticism with the text in the Introduction, Results, and Discussion sections is two-fold. One, there is quite a bit of redundancy through these sections (e.g. definitions of terms) which could be eliminated to streamline the messaging. Two, the authors struggled with use of proper English throughout the paper and should have a primary English speaking editorial assistant proof-read the manuscript during the revision process. This is not meant as an insult, but the unusual phrasing in numerous sentences lowered the readability of the paper in my opinion. I would be happy to offer suggested edits in minor comments in a revised version of this paper. I have highlighted examples of many of these grammatical/syntax issues throughout the marked-up PDF that accompanies my written report.

3. The Discussion section is generally well-composed and was enlightening. Yet, I was left wanting more discussion of suspensory function and how it differs among the taxa represented in this manuscript. The authors preferred to keep the information in the section more pedantic. That said, I did believe that the authors made several convincing arguments about the plasticity of internal structure versus constraints on bone shape among suspensory taxa. I also believe that the authors have not used the available literature/data to the fullest extent. There are several papers that have investigated organ-level and tissue-level properties of long bones in Xenarthrans, and in particular, in extant tree sloths (e.g. Mossor et al. 2022). Sloths are arguably the quintessential obligate suspensory mammals and data contained in these published works could help broaden discussion on convergent versus divergent traits related to suspensory function that I am suggesting that this manuscript is somewhat lacking. I suspect that the authors chose to more strictly focus on structural properties citing that most previous publications focused on organ-level material properties and/or geometric shape variation. Nevertheless, previous studies (including our own work) have noticed irregularities in bone density and porosity in xenarthran (e.g. tree sloth) limb bones. I would personally like to better understand the selection pressures which led to the traits and how they are related or beneficial to sloth limb bone loading.

Reviewer #2 (Remarks to the Author):

I was glad to have the opportunity to review "The macroevolutionary 'common-garden' experiment: evolvability of bone organization levels in repeated acquisitions of slow arboreality in mammals". This

is an important and interesting study that attempts to establish the differing degrees of influence on convergence that are had by bone shape and bone structure – namely, that of the humerus and the femur. This study is of broad interest to a wide group of biologists, including evolutionary biologists, macroevolutionary researchers, and functional morphologists.

The study utilizes geometric morphometrics in its examination of humeral and femoral shape via landmarking, and examines humeral and femoral structure via cross-sections and volumes of interest. Additionally, the study quantifies convergence via use of Stayton's C-indices, more specifically using C1-C3. The results of the study suggest that structural data is far more influential in determining convergence than shape data, although shape data does make some minor contributions to overall convergence. A few concepts discussed could use some justification in either the introduction or the methods section, such as why shape data was quantified using 3D geometric morphometrics, as opposed to other methods, such as 2D geometric morphometrics or functional indices via linear measurements, or why other long limb bones, such as the ulna, radius, tibia, or fibula, were not examined in this study.

However, to me, the biggest issue with this study lies with a recently discovered issue with the C-indices, which are used extensively throughout the study. A preprint (available here: <https://www.biorxiv.org/content/10.1101/2022.10.18.512739v2>) has indicated that, on occasion, the C-indices as currently defined will misinterpret highly divergent taxa as convergent, which can lead to misleading results. While this paper is still in review and has yet to be formally published, the `conevol` R package that contains the C-indices has recently been updated to include the proposed alternative indices that the preprint provides as a potential solution to this problem. I will additionally note that the original author of the C-indices is also a co-author on this preprint. I think this study would greatly benefit from the consideration of the issues pointed out by this preprint. Utilizing the new metrics may or may not substantially change the results of this study, since these problems often occur with taxa that are highly divergent, but a number of previous studies that have used the C-indices, including some cited in this study (i.e. Grossnickle et al. 2020), have had differing results once the new indices were implemented.

Other than this issue, the rest of my comments largely revolve around improving clarity, both linguistic and visual. I have made full comments in the attached document, please see it for more details. As they currently stand, the discussion of the results of this study are sound and supported by the data provided. However, as the C-index results are tied to the vast majority of results for this study, it is difficult to determine if these results will remain sound until the new version of the indices is used in the analyses. The identification that structural traits more frequently differentiate the so-called "slow" and "non-slow" arboreal mammals that shape traits is certainly a compelling result – how this will end up relating to the degrees of convergence of these groups remains to be seen. In short, I believe this is a strong start to a manuscript, and I am eager to see updated results once the new C-indices are used.

Reviewer #3 (Remarks to the Author):

This manuscript addresses the concept of "mosaic evolution" at different spatial scales in bone—shape and material—using convergent locomotor evolution among arboreal mammals as a natural experiment. It tests an interesting hypothesis, and I imagine the results will be useful for the many researchers of morphological evolution and the evolution of bone, especially in mammals. Overall, the manuscript is well-organized, which I appreciate, although frequently the sentence construction is awkward or ambiguous and consequently difficult to read. I recommend the authors undertake another copy edit. 
A main issue might be with the convergence quantification, especially since the results of the experiment hinge on convergence metrics. There are potentially serious issues with the convergence metrics as originally defined by Stayton, such that their computation can lead to misleading/incorrect results. I suggest the authors read closely the following preprint to see if these issues apply:

<https://www.biorxiv.org/content/10.1101/2022.10.18.512739v2> A cautionary note on quantitative measures of phenotypic convergence, Grossnickle et al.

Introduction:

It would be helpful to state why you think bone material parameters might be more evolvable, rather than just focus on why shape might be less evolvable. One might argue that bone shape might evolve in response to ecology because ligaments, tendons, and especially muscles do as well. One might also argue that this same organizational complexity also applies to determining bone material properties. So, the manuscript would benefit from further explanation and references.

2nd paragraph — I wouldn't say mosaic evolution is "arising" as a major driver. Rather, I'd say it is "increasingly recognized" or something similar.

The last line of the introduction that expounds the prediction is particularly confusing. I'd recommend restating.

Results:

Please summarize what measurements you actually took in the text. Tables 1 and 2 are not interpretable unless one already knows what all of the acronyms mean. This is particularly true of the structural variables. Perhaps you can also list the acronym definitions in the table captions? You might be able to expand Fig 1 to further illustrate structural measurements from your VOIs. If the Results section will be printed before Methods, then the reader needs to know what you are actually measuring (and then calculating from those measurements) to understand your Results and Discussion.

Methods:

State and reference what software you used to calculate the trabecular parameters.

We thank the three Reviewers for reading, revising, and suggesting important improvements to our manuscript. Moreover, we thank them for their overall positive evaluation. As detailed below, we believe that we dealt with the concerns raised by Reviewers in detail and we provided substantial justification for our approach.

We thank two of the Reviewers (#2 and #3) for bringing to our attention the recent preprint of Grossnickle et al. We read it carefully, including the updated version (<https://doi.org/10.1101/2022.10.18.512739>; 09th of June 2023; published after the Reviewers provided comments for this manuscript). We explored the novel indices that Grossnickle et al. propose. Following the Reviewers' suggestions we worked to implement the updated C indices (that Grossnickle et al. call Ct indices). For this, the corresponding author had a personal email exchange with David Grossnickle and William H. Brightly (the first two authors of the preprint). We found the preprint and the novel indices proposed by Grossnickle et al. an impressive work that surely will impact future convergence analyses. However, after careful discussion among us we came to the conclusion that the issues in C indices computation highlighted by Grossnickle et al. and the solution that they propose with Ct indices, do not align with our definition of convergent evolution. The initial C indices' definition, on the other hand, align with what many researchers – us included – expect when they seek to quantify convergence. Remarkably, the functions needed to compute C indices are still present in 'convevol 2.0', hence it is still possible to compute C-indices instead of Ct-indices. Indeed, the problems related to C indices that they detect, i.e. C scores inflated for taxa occupying outlying morphologies and convergence detected with not strictly convergent evolutionary trajectories, are not methodological limitations, e.g. programming bugs, that make C indices results meaningless. Rather, these issues just do not align with the working definition of convergence of Grossnickle et al., as they clearly explain in their preprint. We added a dedicated Discussion section ('Convergence measures and definitions', L.317-357, new MS) where we treat in detail this aspect and why C indices are more suitable than Ct indices for our aims. We would like to stress the fact that an agreement on the definition of convergent evolution has still to be achieved.

Across the MS, we did some changes related to this main concern about C-indices:

- To take into account the recently published preprint on the new version of convevol, we cited it in all the sections of the manuscript where we talk about C indices. They are, as mentioned above, computed through functions still implemented in the new version of the R package. Hence, the entire work can be fully reproduced using convevol 2.0.
- In Results, where we first introduce C indices (LL. 119-120, old MS; L. 150-159, new MS), we mention the newly introduced Discussion section 'Convergence measures and definitions'
- Accordingly, we changed the description of C indices and their biological meaning: C indices allow us to have a measure of the amount of morphological changes toward the same region of the morphospace, closer to our concept of convergence and its identification:
 - (L.152-153, new MS): We added '(...) focusing on the magnitude of phenotypic change that leads putatively convergent taxa to evolve similar morphologies (...)'
 - We removed '(...) informs on how much past morphological differences have been reduced by convergence (...)'

- **To be consistent, we do not refer anymore to ‘convergent’ trajectories when we describe phylomorphospaces, but we talk about trajectories that point toward the same distinct region of the morphospaces (as detailed in ‘Convergence measures and definitions’)**
- **L.482-488 (new MS): In Methods, we specify that we adopted the approach of C indices (and we refer to the name of the related functions in ‘convevol 2.0’), instead of Ct indices, and we briefly justify it referring to the definition of convergence (also mentioning the Discussion section ‘Convergence measures and definitions’ for a more detailed treatment)**
- **The Discussion sentences dealing with directions of evolutionary trajectories on phylomorphospaces were adapted and moved to a new section (L. 253-278, new MS), introduced to discuss more in detail slow arboreal lifestyle across the studied taxa and to answer a comment of Reviewer #1 (comment 6, see below).**

Answers to reviewers' comments (highlighted in **bold**):

Reviewer #1	
Reviewer comments	Response
1. Having a strong personal interest in suspensory locomotor structure and function, I was particularly interested in reviewing this manuscript. My overall impression is favorable. I also believe that this work will make a valuable contribution to the current literature on bone structure and function. It certainly provides new perspectives of what levels of bone provide the greatest opportunity to observed important traits that can be evaluated for convergence or divergence. As opposed focusing on numerous minor comments and suggested edits, I will focus my review on major comments for the authors to consider.	We thank Reviewer #1 for providing overall positive feedbacks on our work
2. I am quite impressed with the level care and critical attention to details with data analysis. However, I must admit that I am unfamiliar with several of the R ‘toolkits’ that the authors employed. Nonetheless, I am quite familiar with previous work of two of the authors, and I trust that the analysis was conducted correctly and the statistical methods were also applied correctly.	We thank Reviewer #1 for the positive feedbacks on the thoroughness of our statistical analysis.
3. I think that the tables and figures are useful for condensing the data. Several of them are indeed, well done. If I am to make one criticism here it would be that the size of the panels in Fig. 4 are a bit too small. The 3D-PCA phylomorphospace plots are accordingly difficult to interpret, and the 2D plots are quite garbled.	We thank Reviewer #1 for positive feedback on our tables and figures. We re-structured Fig. 4. We removed 3D phylomorphospaces (and all related text in the manuscript) since, as also pointed out by Reviewer #2 (see below), they do not convey too much additional information and we believe that they would have been difficult to be enlarged without affecting other elements. It allowed us to enlarge the size of PC1-PC2 phylomorphospaces, which now are not garbled anymore.

4. The authors did an excellent job citing referenced literature. In particular, terminology (e.g. mosaic evolution) is clearly defined and well-referenced. My main criticism with the text in the Introduction, Results, and Discussion sections is two-fold. One, there is quite a bit of redundancy through these sections (e.g. definitions of terms) which could be eliminated to streamline the messaging.	We thank Reviewer#1 for positive feedback on our literature referencing and terminology definition. To cope with redundancy through Introduction, Results and Discussion:  - The short definition of convergence (L. 93, old MS), repeated at the beginning of Results, was removed - We refrained from re-defining macroevolutionary common garden experiment (L. 180-182, old MS) and mosaic evolution (L. 182-182, old MS), at the beginning of Discussion - We moved most of the background information to the Introduction (e.g. L. 185-187, old MS; L.65-76, new MS), while Discussion was restructured so that now its beginning is streamlined and directly goes to the discussion of results (L.200-205, new MS)
5. Two, the authors struggled with use of proper English throughout the paper and should have a primary English speaking editorial assistant proof-read the manuscript during the revision process. This is not meant as an insult, but the unusual phrasing in numerous sentences lowered the readability of the paper in my opinion. I would be happy to offer suggested edits in minor comments in a revised version of this paper. I have highlighted examples of many of these grammatical/syntax issues throughout the marked-up PDF that accompanies my written report.	We thank the Reviewer for the multiple annotations. We have taken them into consideration and thoroughly revised the manuscript to polish the English (see section on Grammatical/syntax issues below)
6. The Discussion section is generally well-composed and was enlightening. Yet, I was left wanting more discussion of suspensory function and how it differs among the taxa represented in this manuscript. The authors preferred to keep the information in the section more pedantic. That said, I did believe that the authors made several convincing arguments about the plasticity of internal structure versus constraints on bone shape among suspensory taxa. I also believe that the authors have not used the available literature/data to the fullest extent. There are several papers that have investigated organ-level and tissue-level properties of long bones in Xenarthrans, and in particular, in extant tree sloths (e.g. Mossor et al. 2022). Sloths are arguably the quintessential obligate suspensory mammals and data contained in these published works could help broaden discussion on convergent versus divergent traits related to suspensory function that I am suggesting that this manuscript is somewhat lacking. I suspect that the authors chose to more strictly focus on structural properties citing that most previous publications focused on organ-level material properties and/or geometric shape variation. Nevertheless, previous studies (including our own work) have noticed irregularities in bone density and porosity in xenarthran (e.g. tree sloth) limb bones. I would personally like to better understand the selection pressures which led to the traits and how they are related or beneficial to sloth limb bone loading.	We thank Reviewer #1 for the positive feedback on the Discussion. We introduced a new sub-section (L.253-278, new MS) in which we detail how lifestyles of the taxa differ. We recognize that the main contributors to convergence are suspensory slow arboreal mammals. We thank Reviewer #1 for literature suggestion. Mossor et al. 2022 is a very nice work and it is now cited in the newly introduced section. That being said, we believe that focusing more on xenarthrans and/or suspensory locomotion would be outside the focus of this particular work. Although it is true that sloths are likely the quintessential obligate suspensory mammals, they are only 2/7 of the taxa that we study in this work. We are afraid that detailing their functional morphology as Reviewer #1 suggests, would be hard to justify in the absence of similar treatment for other slow arboreal mammals. Also, not all the slow arboreal taxa are suspensory since suspensory locomotion is not part of the definition of slow arboreal that we use in this work (L. 82-84, new

	MS). We are aware of the peculiar bone density and porosity in xenarthrans (e.g. Alfieri et al. 2021) and the corresponding author had a conversation with A. Mossor on it, during a conference. Moreover, a detailed functional morphological and evolutionary analysis focusing solely on slow arboreal xenarthrans was recently published (e.g. Alfieri et al. 2022).
Grammatical/syntax issues highlighted by Reviewer #1 on the PDF version of the MS	
L. 18 (old MS): ‘It allows to’	L. 17 (new MS): ‘researchers’ was added after ‘allows’
L. 29 (old MS): ‘clearly confirm our assumption, too’	L. 24 (new MS): Changed to ‘we found confirmation for our assumption’ in a sentence that was, however, changed
L. 37 (old MS): ‘(...) the differential (phenotypic) ability to (...) respond to opportunity (...)’.	L. 33-34 (new MS): we replaced the original phrasing from Jablonski, 2022 with a similar definition, (i.e. the propensity to evolve in response to opportunity)
L. 55-56 (old MS): ‘studies compared’	L. 55-56 (new MS): Changed to ‘... , but no macroevolutionary studies have attempted to compare evolvability pattern of bone shape with that of bone structure ...’
L. 76 (old MS): ‘Following the aforementioned’	Entire sentence deleted
L. 102 (old MS): ‘set apart’	L. 132 (new MS): Changed to ‘distinguished’
L. 137 (old MS): ‘by the femoral data, too’	L. 167 (new MS): Changed to ‘is also mirrored by the femoral data’s distribution’
L. 162 (old MS): ‘is preponderantly driven’	Deleted
L. 179 (old MS): ‘Following Jablonski’	Deleted
L. 181 (old MS): ‘partly keep constant’	The sentence is not present in the new MS version
L. 182 (old MS): ‘and allow to gain insight’	The sentence is not present in the new MS version
L. 198 (old MS): ‘a same sample in a dedicated study’	L. 74 (new MS): we replaced it with ‘a homogeneous sample’
L. 200 (old MS): ‘ca. 170-million years-old clade’	The sentence is not present in the new MS version
L. 202 (old MS): ‘and are more in most cases’	L. 211 (new MS): Changed to ‘and are, in most cases, ..’
L. 211 (old MS): ‘preponderantly contribute’	L. 215 (new MS): we replaced it with ‘mainly contribute’
L. 212 (old MS): ‘All results point to the fact that’	L. 216 (new MS): Changed to ‘All results are in agreement with the expectation that’
L. 222 (old MS): ‘convergently freed from’	L. 226 (new MS): Changed to ‘freed from these constrains, convergently’
L. 224 (old MS): ‘but both traits proved’	L. 228-229 (new MS): Changed to ‘but both traits have been found to reflect ecology’
L. 249 (old MS): ‘with the fact that’	L. 281 (new MS): Changed to ‘may be a factor explaining that’
L. 254 (old MS): ‘is the ontogenetic one’	L. 287 (new MS): we replaced ‘the ontogenetic one’ with ‘ontogeny’
L. 264 (old MS): ‘vary relatively to one another’	Deleted

L. 265 (old MS): ‘frequently tackled’	L. 297 (new MS): Changed to ‘often studied using the concepts of’
L. 268 (old MS): ‘relatively to other’	L. 300 (new MS): Changed to ‘relative to other’
L. 275 (old MS): ‘tempo and mode’	Deleted
L. 279 (old MS): ‘they might have’	L.310 (new MS): Changed to ‘they could have’
L. 282-283 (old MS): ‘In support of this’	L. 313-314 (new MS): Changed to ‘In support of this assumption’
L. 352 (old MS): ‘run in this work are two-sided’	L. 439 (new MS): Changed to ‘tests used in this work are two-sided’
L. 356-357 (old MS): ‘and body mass was used as a covariate. As body mass was not available’	L. 445 (new MS): Changed to ‘and a body size proxy was used as a covariate. As body size proxy, we took the natural’
L. 361-362 (old MS): ‘the effects of lifestyle and body mass’	L. 449-450 (new MS): ‘body mass’ changed to ‘body size’
L. 377 (old MS): ‘warping meshes to maximum and minimum values’	L. 465 (new MS): Changed to ‘warping meshes to maximum and minimum PC score values’
L. 392 (old MS): ‘datasets allows to compare how strongly converge slow arboreal mammals’	L. 480 (new MS): Changed to ‘datasets allows us to compare how strongly converge slow arboreal mammals’
L. 401 (old MS): ‘since it more probably results’	L. 492 (new MS): Changed to ‘since it results more probably’
L. 408 (old MS): ‘We used’	L. 499 (new MS): ‘..Changed to ‘We used the approach of the convol functions...’
L. 420-421 (old MS): ‘Additional information on the extraction of morphological data, PGLSs/ANCOVAs and convergence strength measurements is provided in Supplementary Information (Supplementary Notes 5-8)’	L. 510-511 (new MS): In the new MS version the sentence is not in a dedicated paragraph
Reviewer #2	
Reviewer comments	Response
1. I was glad to have the opportunity to review “The macroevolutionary ‘common-garden’ experiment: evolvability of bone organization levels in repeated acquisitions of slow arboreality in mammals”. This is an important and interesting study that attempts to establish the differing degrees of influence on convergence that are had by bone shape and bone structure – namely, that of the humerus and the femur. This study is of broad interest to a wide group of biologists, including evolutionary biologists, macroevolutionary researchers, and functional morphologists. The study utilizes geometric morphometrics in its examination of humeral and femoral shape via landmarking, and examines humeral and femoral structure via cross-sections and volumes of interest. Additionally, the study quantifies convergence via use of Stayton’s C-indices, more specifically using C1-C3. The results of the study suggest that structural data is far more influential in determining convergence than shape data, although shape data does make some minor contributions to overall convergence.	We thank Reviewer #2 for the positive feedback
2. A few concepts discussed could use some justification in either the introduction or the methods section, such as why shape data was quantified using 3D geometric morphometrics,	L. 388-400 (new MS): In the methods section, we added a justification for our choice to use high-density 3D GM, instead of

as opposed to other methods, such as 2D geometric morphometrics or functional indices via linear measurements,	linear/angle measurements, functional indices, to quantify bone shape
3. or why other long limb bones, such as the ulna, radius, tibia, or fibula, were not examined in this study	We examined the humerus and the femur because an abundant literature is dedicated to these two bones, concerning the anatomical levels and structural sub-regions here addressed and their relationships with ecology. Moreover, we wanted to represent both forelimbs and hindlimbs. It is now made clear with a sentence in the Introduction (L. 87-90, new MS).
4. However, to me, the biggest issue with this study lies with a recently discovered issue with the C-indices, which are used extensively throughout the study. A preprint (available here: https://www.biorxiv.org/content/10.1101/2022.10.18.512739v2) has indicated that, on occasion, the C-indices as currently defined will misinterpret highly divergent taxa as convergent, which can lead to misleading results. While this paper is still in review and has yet to be formally published, the convevol R package that contains the C-indices has recently been updated to include the proposed alternative indices that the preprint provides as a potential solution to this problem. I will additionally note that the original author of the C-indices is also a co-author on this preprint. I think this study would greatly benefit from the consideration of the issues pointed out by this preprint. Utilizing the new metrics may or may not substantially change the results of this study, since these problems often occur with taxa that are highly divergent, but a number of previous studies that have used the C-indices, including some cited in this study (i.e. Grossnickle et al. 2020), have had differing results once the new indices were implemented.	Please see our answer on the main concern of C indices at the beginning of the rebuttal letter
5. Other than this issue, the rest of my comments largely revolve around improving clarity, both linguistic and visual. I have made full comments in the attached document, please see it for more details.	All the edits suggested by Reviewer #2 that are related to punctuation and grammar were accepted and integrated in the new version of the MS, with the exception of ‘L. 46 (original MS): suggested to change ‘is arising’ to ‘arises’. In this case, we preferred to change ‘is arising’ to ‘is increasingly recognised’ [L. 43 (new MS)] following the suggestion of Reviewer #3 (see Reviewer #3’s comment 4, below)
6. As they currently stand, the discussion of the results of this study are sound and supported by the data provided. However, as the C-index results are tied to the vast majority of results for this study, it is difficult to determine if these results will remain sound until the new version of the indices is used in the analyses. The identification that structural traits more frequently differentiate the so-called “slow” and “non-slow” arboreal mammals that shape traits is certainly a compelling result – how this will end up relating to the degrees of	Please see our answer on the main concern of C indices at the beginning of the rebuttal letter

convergence of these groups remains to be seen. In short, I believe this is a strong start to a manuscript, and I am eager to see updated results once the new C-indices are used.	
Other comments made by Reviewer #2 on the PDF version of the manuscript	
L. 18 (original MS): ‘allow’ → There is a missing word here (perhaps "researchers" or "evolution"?), which seems critical to the meaning of this sentence.	L. 17 (new MS): ‘researchers’ was added after ‘allow’
L. 39 (original MS): ‘allow’ → Allows what? Another missing word that is critical to the meaning of the sentence.	L. 35 (new MS): Changed to we are able to’.
L. 52 (original MS): Move "(‘shape’ hereafter)" to just after "External gross morphology". This will make it clearer that 'shape' refers not to the level, but to the morphology, and follows the sentence structure you use when you introduce 'structure'.	We thank Reviewer #2 for the suggestion that we followed (L. 52, new MS):
L. 54 (original MS): Citation(s) needed here in reference to the "recent ... technological advances"	L. 54 (new MS): We added a citation to a recent review (Keklikoglou et al. 2021), summarizing imaging techniques used in biological studies, since the technological advances we refer to are techniques as microCT, allowing to non-destructively access internal structure.
L. 62 (original MS): ‘i.e. shape is less evolvable’: Are you stating this as fact or proposing an interpretation of what the previous stated facts might result in? I suspect it is the latter, in which case you should make this more clear.	L. 64 (new MS): Indeed, the lower evolvability of shape is an expectation deriving from previous evidence. Hence, the sentence was changed to ‘i.e., shape is expected to be less evolvable’.
L. 70 (original MS): How are you defining "arboreal" in this study? Does it include putatively "scansorial" taxa as well, or are you operating under a stricter definition? Worth clarifying	Arboreal taxa in this study are defined as fully arboreal, i.e. spending their time almost entirely on trees. Hence, scansorial taxa are not included. We clarified, reformulating the definition of slow arboreal mammals in LL. 82-84 (new MS) and including “(...) These taxa spend most of their life in trees (...)”. It is added to the rest of the definition that was positively evaluated by Reviewer #2 [(LL. 71-72 (original MS): Good definition of "slow")]
L. 96 (original MS): Be specific, what does "they" refer to?	They referred to ‘Traits deserving further analyses of convergence’. It is now specified (L. 125-126, new MS) adding ‘These traits’.
L. 120 (original MS): Okay, let's talk about the C indices. There is a preprint currently out that indicates that there is a methodological issue with these indices where it will on occasion interpret highly divergent taxa as convergent, which can lead to misleading results. While that paper is still in review, they do provide a solution, and it is worth both being	Please see our answer on the main concern of C indices at the beginning of the rebuttal letter

aware of this issue and either addressing why such issues are not relevant here or utilizing the recently updated <code>conevol v2.0 R</code> package with the new metrics to avoid this issue. The authors should be aware that the issues raised particularly affect C1-C4, three of which are the exact indices they have opted to use. The preprint can be found here: https://www.biorxiv.org/content/10.1101/2022.10.18.512739v2.	
L. 180 (original MS): ‘macroevolutionary common-garden experiment’ → You refer to this terminology here, in the abstract, and in the title of the paper, but it is never referred to again after it is properly introduced here. Why introduce it if you’re not going to use it or contextualize your results with it?	We recalled the main aim of the work, i.e. conducting a macroevolutionary common garden experiment, at the end of Introduction, (L. 91, new MS), and the beginning of Results sections (L. 103, new MS). Still in the Results we reminded that identifying distinct convergence patterns between anatomical levels is the key aspect of the natural experiment that we study (L. 119-121, new MS)
L. 190-201 (original MS): This paragraph is good setup for why you are conducting this study - perhaps it belongs in the introduction?	We followed Reviewer #2’s suggestion and we moved the paragraph to Introduction, adapting it (L. 65-76, new MS)
L. 202-213 (original MS): If these findings hold true after implementing the updated C-indices, they would be some very cool results!	Please see the newly introduced section ‘Convergence measures and definitions’ for our explanation of why we used C-indices instead of Ct-indices.
L. 227 (original MS): If you suspect that the taxa you examined may actually be divergent, that makes the issue regarding the C-metrics I mentioned earlier all the more important to consider.	We are afraid that in the previous version of the manuscript we were not clear in this sentence. Our intention was to outline which are the alternative hypotheses, if one deals with non-convergent traits: they may result from either ‘divergence’ or ‘phylogenetic conservatism’. Since our hypothesis on weak evolvability and convergence of shape was based on higher constraints, we think that it should result in phylogenetic conservatism. Hence, divergence was here mentioned only as a possible alternative explanation to convergence, but it is not the focus of the section. Please see the newly introduced section ‘Convergence measures and definitions’ for our explanation of why we used C-indices instead of Ct-indices.
L. 314 (original MS): For the shape data, why did you opt for using geometric morphometrics, as opposed to, say, linear measurements or functional indices? A justification for why you chose your particular methodologies should either be here or in the introduction	L. 388-400 (new MS): In the methods section, we added a justification for our choice to use high-density 3D GM, instead of linear/angle measurements, functional indices, to quantify bone shape
Fig. 1: I like this figure - it's very distinct in the comparison between shape and structure and makes clear what you mean by each term, which is critical for the rest of the paper.	We thank Reviewer #2 for the positive feedback on our Figure.
Fig. 2: By "time-tree" do you mean that this is a time-calibrated tree? If so, a scale for branch lengths might be useful to include.	Reviewer #2 is right, by ‘time-tree’ we mean that this is a time-calibrated tree. Following Reviewer #2’s suggestion a time scale

	was added and the caption accordingly modified (see Edited Figures below)
Fig. 3: I really like this visual display - it's quite striking! Any reason you have chosen to only make this plot for C2, and not C1 or C3?	We thank Reviewer #2 for the positive feedback on our Figure. The reason why we represent results for only C2 (and not for C1 or C3) is that C2 is the only index for which both shape and most of the structural datasets yielded a significant p-value. Hence, only C2 allows us to make a direct comparison of informative C-values between shape and structure. We now specify this aspect in Fig.3 caption.
Fig. 4: How much visual information are you gaining by displaying a 3D phylomorphospace versus having two 2D phylomorphospaces (one comparing PC1 and PC2, as you already have, and one comparing PC1 and PC3)? I have always found 3D phylomorphospace plots to be more difficult to tease apart, and doesn't add significantly more to the figure. But this is a matter of personal preference.	We changed Fig. 4, only including two 2D PC1-PC2 phylomorphospaces and removing 3D phylomorphospaces. It allowed us to have larger 2D phylomorphospaces, also related to a comment of Reviewer #1 (see comment 3; above). See more details on Fig. 4 changes in the Edited Figures section below.
Fig. 4. The animal images in this plot are a bit tricky to make out - perhaps silhouettes would be clearer?	We followed Reviewer #2's suggestion and we replaced animal images with silhouettes
Extended Data Fig. 1: The text in this geologic time scale is tiny! I can barely read it. With the amount of space you have in each box, it should be trivial to increase the text size to be readable.	We increased the text size of the geological time scale labels
Extended Data Fig. 2: The text in this figure is also quite small and hard to read - consider increasing the size here too!	We increased the size of species labels, numbers in the x-axes and x-axes labels. Moreover, we removed the 'Slow Arboreal' and 'Non-Slow Arboreal' labels on the y-axes, introducing a color legend
Extended Data Fig. 3: Ditto to the previous comment.	We increased the size of species labels, numbers in the x-axes and x-axes labels. Moreover, we removed the 'Slow Arboreal' and 'Non-Slow Arboreal' labels on the y-axes, introducing a color legend
Extended Data Fig. 4: Compared to the previous plots, I think the use of silhouettes here makes the taxa comparisons far more clear to distinguish and greatly improves the plot.	We thank Reviewer #2 for the positive feedback on this plot. Related to this, Figure 4 now includes silhouettes (see above)
Extended Data Fig. 4: The clearest parts of this PCA plot are the blue dots. By saying that the non-slow arboreal mammals are shown in blue, these dots are the first thing I connect that part of the sentence to. The convex hulls are blurry and hard to make out, and the low opacity makes it difficult to connect those to the colors mentioned. Hard borders to the convex hulls and more distinct colors would help with this issue.	We changed blue dots into black dots to avoid confusion with the blue region of non-slow arboreal mammals. Moreover, for convex hulls we do not use blurry colors and we added a border.
Extended Data Fig. 5: Ditto to the previous comment.	We changed blue dots into black dots to avoid confusion with the blue region of non-slow arboreal mammals. Moreover, for convex hulls we do not use blurry colors and we added a border.
Reviewer #3	
Reviewer comments	Response

1. This manuscript addresses the concept of “mosaic evolution” at different spatial scales in bone—shape and material—using convergent locomotor evolution among arboreal mammals as a natural experiment. It tests an interesting hypothesis, and I imagine the results will be useful for the many researchers of morphological evolution and the evolution of bone, especially in mammals. Overall, the manuscript is well-organized, which I appreciate, although frequently the sentence construction is awkward or ambiguous and consequently difficult to read. I recommend the authors undertake another copy edit	We thank Reviewer #3 for the positive feedback. As for sentence construction...
2. A main issue might be with the convergence quantification, especially since the results of the experiment hinge on convergence metrics. There are potentially serious issues with the convergence metrics as originally defined by Stayton, such that their computation can lead to misleading/incorrect results. I suggest the authors read closely the following preprint to see if these issues apply: https://www.biorxiv.org/content/10.1101/2022.10.18.512739v2 A cautionary note on quantitative measures of phenotypic convergence, Grossnickle et al	Please see the newly introduced section ‘Convergence measures and definitions’ for our explanation of why we used C-indices instead of Ct-indices.
3. Introduction: It would be helpful to state why you think bone material parameters might be more evolvable, rather than just focus on why shape might be less evolvable. One might argue that bone shape might evolve in response to ecology because ligaments, tendons, and especially muscles do as well. One might also argue that this same organizational complexity also applies to determining bone material properties. So, the manuscript would benefit from further explanation and references.	LL. 61-62 (new MS): We added a sentence (with references) [“(…) On the other hand... mechanical stresses (...)”] that justifies why we think that bone structure may be more evolvable than bone shape. As for the possibility that ligaments, tendons, and especially muscles would evolve according to large shape modifications, it is not impossible, of course. However, we think that it is generally less likely due to the complex network of interactions and potential consequences deriving from large modifications (hence we added ‘and potentially deleterious’ to ‘extensive’, when talking about ‘reorganisation of many interacting structures’) (L. 60 (new MS). You may read the Introduction of Kivell 2016 (that we cite in the MS), for a more detailed treatment of shape vs. structure expected ecological adaptation
4. 2nd paragraph — I wouldn’t say mosaic evolution is “arising” as a major driver. Rather, I’d say it is “increasingly recognized” or something similar.	L. 43 (new MS): ‘arising’ was changed to ‘increasingly recognised’
5. The last line of the introduction that expounds the prediction is particularly confusing. I’d recommend restating.	The last line of the Introduction in the old MS version was removed and replaced with a short paragraph in which we summarize main results and our conclusions (to meet a formatting guideline of this journal) (L. 91-96, new MS)
6. Results: Please summarize what measurements you actually took in the text. Tables 1 and 2 are not interpretable unless one already knows what all of the acronyms mean. This is particularly true of the structural variables. Perhaps you can also list the acronym definitions in the table captions? You might be able to expand Fig 1 to further illustrate structural measurements from your VOIs. If the Results section will be printed before Methods, then the reader needs	As suggested by Reviewer #3, in the main text (i.e. beginning of Results, where we first mention the extraction of shape and structure data, L. 105-121, new MS), we now specify where 3D GM PCs come from and we list the cross-sectional properties and trabecular parameters that we extracted and analyzed (both extended name and acronyms). Moreover, we follow Reviewer #3’s

to know what you are actually measuring (and then calculating from those measurements) to understand your Results and Discussion.	suggestion also in listing the traits acronyms and definitions in Table 1-2 captions. Although including a description of structural measurements in Fig. 1 would also increase the understanding of structural traits, we however prefer to leave Fig. 1 as it is now, since we believe that the figure should not include too many details and should focus, instead, on main differences between the level ‘shape’ and the level ‘structure’. In this regard, Reviewer #2 provided a positive feedback on this aspect (“I like this figure - it's very distinct in the comparison between shape and structure and makes clear what you mean by each term, which is critical for the rest of the paper”).
7. Methods: State and reference what software you used to calculate the trabecular parameters.	Trabecular parameters were computed through the FIJI plugin BoneJ and it is now specified and referenced: ‘In FIJI, from each VOI we extracted six trabecular parameters through the respective routines of the plugin BoneJ (ref)’ (L. 416, new MS).

Edited Figures

Fig. 2. The seven independent acquisitions of the slow arboreal lifestyle (red sections), reconstructed with Stochastic Character Mapping, are shown on the time-calibrated tree of the taxa for which morphological data were obtained (see Extended Data Fig. 1 for an extended version). A million-years ago (Mya) time scale is present.

Changes:

Following the Reviewer #2's suggestion, in this new version of Figure 2 a time scale (with unit 'Million Years Ago', abbreviation: Mya) was added and the caption was accordingly modified.

Fig. 3. C2 values resulting from convergence analyses performed through ‘convevol’^{35,36}, on humeral (above) and femoral (below) datasets. C2 is the only index for which results are shown in the figure, since it is the only for which shape and most of structural datasets yielded significant p-values (<0.05), hence returning informative values. Schemes A-C refer to convergence analyses repeated to account for potential biases (as detailed in the Methods). Results for the femoral medial condylar structural dataset and the Scheme B analysis of the femoral average diaphyseal structural level are not presented here because they did not yield significant p-values for C2 tests (Extended Data Table 3).

Changes:

Following a comment of Reviewer #2, in the caption we now make clear why C2 is the only index for which we show results in the Figure

Fig. 4. Humeral (above) and femoral (below) convergence in slow arboreal mammals, shown in phylomorphospaces. Taxa positions reflect morphology: closer taxa are more similar to each other. The relatively distinct position of slow arboreal mammals (light red) compared to non-slow arboreal mammals (blue) demonstrates that they tend to resemble each other. Phylomorphospaces (left panel) allow to reconstruct trajectories of phenotypic evolution (here they are labelled A. three-toed sloths, B. two-toed sloths, C. koala, D. koala lemurs, E. lorisisds, F. silky anteater, G. sloth lemurs). As expected for convergent taxa, several slow arboreal mammals occupy a distinct region of phylomorphospaces through converging trajectories (red arrows). In variable loadings plots (right panel), traits mostly contributing to slow arboreal mammal convergence (orange) are identified evaluating their vectors' angles (informing on the trait-PC_{pms} correlation strength), directions (telling if the trait-PC_{pms} correlation is positive or negative) and lengths (reflecting how the trait contributes to taxa distribution on the plot). PC1_{pms}-PC3_{pms} and PC2_{pms}-PC3_{pms} biplots and related variable loadings plots are provided in Extended Data Figs. 6-7

Changes:

We removed 3D phylomorphospaces as suggested by Reviewer #2 (and all the parts of text referring to 3D phylomorphospace), and now we only have two 2D PC1-PC2 phylomorphospaces. It allowed us to have larger plots, as suggested by Reviewer #1. We think that for the aim of this work, it is particularly important to highlight that slow arboreal mammals tend to occupy the same region of the morphospace through related evolutionary trajectories, (i.e. they morphologically converge) and that internal structural traits more strongly contribute to this pattern (i.e. structure is more evolvable). Hence, we believe that PC1-PC2 2D phylomorphospaces + variable loading plots are sufficient. PC1-PC3 and PC2-PC3 2D phylomorphospaces are not crucial for a main text Figure (they are in Extended Data, Extended Data Fig. 6-7). Also, following a comment of Reviewer #2, we replaced animal images with silhouettes.

Extended Data Fig. 1. Time-calibrated phylogeny used to perform Ancestral Lifestyle Reconstruction. Twelve additional taxa (not shown in Fig. 2) were included. On pie charts on internal nodes the most likely ancestral lifestyle reconstruction is showed.

Changes: Following a suggestion of Reviewer #2 we increased the size of the geological time scale labels

Extended Data Fig. 2. Boxplots with the distribution of mean taxa results for humeral traits that yielded a significant relationship with slow arboreal ecology through PGLSs and ANCOVAs. Structural traits showing a significant correlation with body mass too are shown with size-corrected values. Each box starts with 1st quartile, ends with the 3rd quartile and contains the mean (shown with a black triangle) and the median (shown with a vertical line). Whiskers indicate minimum and maximum values.

Changes: Following a suggestion of Reviewer #2, we increased the size of species labels, numbers in the x-axes and x-axes labels. Moreover, we removed the ‘Slow Arboreal’ and ‘Non-Slow Arboreal’ labels on the y-axes, introducing a color legend. Since due to the increased size of species labels linking names to the corresponding dots became challenging, we introduced black lines that connect names to dots

Extended Data Fig. 3. Boxplots with the distribution of mean taxa results for femoral traits that yielded a significant relationship with slow arboreal ecology through PGLSs and ANCOVAs. Structural traits showing a significant correlation with body mass too are shown with size-corrected values. Each box starts with 1st quartile, ends with the 3rd quartile and contains the mean (shown with a black triangle) and the median (shown with a vertical line). Whiskers indicate minimum and maximum values.

Changes: Following a suggestion of Reviewer #2, we increased the size of species labels, numbers in the x-axes and x-axes labels. Moreover, we removed the ‘Slow Arboreal’ and ‘Non-Slow Arboreal’ labels on the y-axes, introducing a color legend. Since, due to the increased size of species labels, linking names to the corresponding dots became challenging, we introduced black lines that connect names to dots

Extended Data Fig. 4. Left panel: Humeral shape variability, highlighted through a 3DGM PC1-PC2 biplot. Slow arboreal mammals are shown in red (together with silhouettes for the seven main clades) while non slow arboreal mammals are shown in blue. Right panel: the main shape variability captured by PC1 and PC2 is shown through maximum/minimum PC1 and PC2 scores warped on the mesh of *Eucholoeops* sp. FMNH P13280 (additionally inflated in the diaphyseal region and smoothed to delete surface micro-cracks, to optimise the warping process).

Changes: Following a suggestion of Reviewer #2, we changed blue dots into black dots to avoid confusion with the blue region of non-slow arboreal mammals. Moreover, for convex hulls we do not use blurry colors and we added a border. We also increased the species names size.

Extended Data Fig. 5. Left panel: Femoral shape variability, highlighted through a 3DGM PC1-PC2 biplot. Slow arboreal mammals are shown in red (together with silhouettes for the seven main clades) while non slow arboreal mammals are shown in blue. Right panel: the main shape variability captured by PC1 and PC2 is shown through maximum/minimum PC1 and PC2 scores warped on the mesh of *Bradypus* sp. ZMB Mam-33806 (additionally inflated in the diaphyseal region to optimise the warping process).

Changes: Following a suggestion of Reviewer #2, we changed blue dots into black dots to avoid confusion with the blue region of non-slow arboreal mammals. Moreover, for convex hulls we do not use blurry colors and we added a border. We also increased the species names size.

Bibliography

- Alfieri, F., L. Botton-Divet, J. A. Nyakatura, and E. Amson. 2022. Integrative approach uncovers new patterns of ecomorphological convergence in slow arboreal xenarthrans. *J Mamm Evol*, doi: 10.1007/s10914-021-09590-5.
- Alfieri, F., J. A. Nyakatura, and E. Amson. 2021. Evolution of bone cortical compactness in slow arboreal mammals. *Evolution* 75:542–554.
- Grossnickle, D. M., W. H. Brightly, L. N. Weaver, K. E. Stanchak, R. A. Roston, S. K. Pevsner, C. T. Stayton, P. D. Polly, and C. J. Law. 2023. Challenges and advances in methods for measuring phenotypic convergence. Pre-print. bioRxiv, 2023-06-09.
<https://doi.org/10.1101/2022.10.18.512739>
- Keklikoglou, K., C. Arvanitidis, G. Chatzigeorgiou, E. Chatzinikolaou, E. Karagiannidis, T. Koletsa, A. Magoulas, K. Makris, G. Mavrothalassitis, E.-D. Papanagnou, A. S. Papazoglou, C. Pavloudi, I. P. Trougakos, K. Vasileiadou, and A. Vogiatzi. 2021. Micro-CT for biological and biomedical studies: a comparison of imaging techniques. *Journal of Imaging* 7:172.
- Kivell, T. L. 2016. A review of trabecular bone functional adaptation: what have we learned from trabecular analyses in extant hominoids and what can we apply to fossils? *J. Anat.* 228:569–594.
- Mossor, A. M., J. W. Young, and M. T. Butcher. 2022. Does a suspensory lifestyle result in increased tensile strength? Organ-level material properties of sloth limb bones. *Journal of Experimental Biology* 225:jeb242866.

REVIEWERS' COMMENTS:

Reviewer #2 (Remarks to the Author):

Thank you for the opportunity to review the revised manuscript for this project. The authors have undertaken substantial revisions, and have addressed essentially all of my initial concerns. The new section of the discussion entitled "Convergence measures and definitions" is a welcome addition, and clarifies the difference in convergence definitions between Grossnickle et al. and the present work. These differences in definition I think are sensible considering the focus and aims of this work, and are a solid justification for continuing to use the C-indices over the Ct-indices. My only suggestion regarding the convergence definition is to include an explicit definition for convergence as it is used in this study in the introduction, so that readers do not have to wait until the discussion, where it is currently defined, to determine how the authors are treating convergence. In my last review, I stated that the conclusions were sound and well-supported by the data so long as a justification for using the C-indices over the Ct-indices was found, and now that such justification has been satisfactorily provided, I stand by that statement.

There are additionally many improvements that have been made regarding the clarity of the text and figures of the manuscript, and it now reads much more cleanly. The added justifications throughout the text strengthen the reasoning behind the choices made here, and overall the manuscript is far more robust this time. There are no obvious parts of the manuscript that require drastic improvement, and as it stands the work is comprehensible and approachable.

Overall, this revision is a significant improvement of the manuscript, and I no longer have any major concerns with the work as it stands now. This study remains of interest to a broad range of biologists, and will be a valuable contribution to the field. I look forward to seeing the work that continues to come from these authors in the future.

Reviewer #3 (Remarks to the Author):

I appreciate the authors' work on this version of the manuscript. I am not quite sure about their "... response to my suggestion that they undertake another copy edit, when the other two reviewers clearly agreed. I am grateful that the other reviewers undertook this copy edit with their many suggestions for the authors, as the manuscript is much improved.

My comments below are meant to strongly urge the authors to consider their use of the term "evolvability" and how they describe their views on convergence, and to make sure their text aligns with their own thoughts. That is, I think I may disagree, but I don't have any specific edits to suggest, and I don't think it is too terrible for the authors to opine as they wish.

I appreciate that the authors better justified their shape vs structure argument. I think they may have gone a bit rough on shape analyses, though. Certainly there is much we have learned about shape correspondence with functional regime. And, there are certainly cases of muscles, ligaments, etc. massively changing (e.g., bats; Bahlman et al. 2017 J Anat). It also seems odd that there is no mention of the plasticity (i.e., within an individual's lifetime) that is inherent in the argument the authors are making (and is the focus of the Kivell intro they suggested). Plasticity may be linked to evolvability, or it may inhibit evolvability---in both cases you might expect plastic traits to align with function in the way you tested in this study.

My other issue is with the added section in response to Grossnickle et al.:

If the authors wish to use a large portion of the Discussion to critique the Grossnickle et al. findings (which is fine), it would be helpful if they included a demonstrative figure illustrating what is on lines 337-346. As written, it is a bit ambiguous. For instance, what exactly does this mean: "if the distance between the corresponding reconstructed ancestral trait values happen to be longer than the observed values of focal lineages."? Do you mean a comparison between the distance(s) between, say, ancestral state values at nodes and the distances between tips? Or the values at the tips?

I also don't quite understand -- isn't it necessary, for taxa to have "converged", for them to have ancestors that were more "divergent"? After all, sister taxa that all cluster together would not be considered convergent. In my experience the term is only used--either technically or colloquially--when considering taxa from different parts of a phylogeny, interspersed at the tips with taxa in different functional and morphological regimes (if considering functional-morphological convergence). I am not sure the authors mean to suggest anything different, but this is how I read the paragraph, so they may want to consider clarifying.

Answers to reviewers' comments:

Reviewer #2	
Reviewer comments	Response
Thank you for the opportunity to review the revised manuscript for this project. The authors have undertaken substantial revisions, and have addressed essentially all of my initial concerns. The new section of the discussion entitled “Convergence measures and definitions” is a welcome addition, and clarifies the difference in convergence definitions between Grossnickle et al. and the present work. These differences in definition I think are sensible considering the focus and aims of this work, and are a solid justification for continuing to use the C-indices over the Ct-indices. My only suggestion regarding the convergence definition is to include an explicit definition for convergence as it is used in this study in the introduction, so that readers do not have to wait until the discussion, where it is currently defined, to determine how the authors are treating convergence. In my last review, I stated that the conclusions were sound and well-supported by the data so long as a justification for using the C-indices over the Ct-indices was found, and now that such justification has been satisfactorily provided, I stand by that statement. There are additionally many improvements that have been made regarding the clarity of the text and figures of the manuscript, and it now reads much more cleanly. The added justifications throughout the text strengthen the reasoning behind the choices made here, and overall the manuscript is far more robust this time. There are no obvious parts of the manuscript that require drastic improvement, and as it stands the work is comprehensible and approachable. Overall, this revision is a significant improvement of the manuscript, and I no longer have any major concerns with the work as it stands now. This study remains of interest to a broad range of biologists, and will be a valuable contribution to the field. I look forward to seeing the work that continues to come from these authors in the future.	We thank Reviewer #2 for providing positive feedback on our revised work and we are glad to know that our revised version of the manuscript addresses Reviewer #2’s initial concerns. Following Reviewer #2’s suggestion, we added a new Introduction paragraph (LL. 44-52) in which we briefly introduce the issue of convergence indices and related definitions. We emphasize our position concerning ‘strictly geometry-based definitions’ on which Grossnickle et al.’s Ct-indices are based. We briefly outline that our approach focuses more on other indicators of convergence (i.e. convergent taxa that are set apart from closely related taxa). We added ‘As detailed in Discussion’ to make the reader understand that we will further elaborate on this aspect in the Discussion section. Since both ‘geometry-based’ definitions and our approach rely on criteria related to phylomorphospaces, in (LL. 41-42) we briefly introduce the concept of phylomorphospace in the previous paragraph: ‘(...) arising from the study of ecologically convergent lineages (...)’, becomes ‘(...) often arising from the study of ecologically convergent lineages on phylomorphospaces (...)’.
Reviewer #3	
Reviewer comments	Response
I appreciate the authors' work on this version of the manuscript. I am not quite sure about their "...” response to my suggestion that they undertake another copy edit, when the other two reviewers clearly agreed. I am grateful that the other reviewers undertook this copy edit with their many suggestions for the authors, as the manuscript is much improved.	We thank Reviewer #3 for providing positive feedback on this new version of our manuscript. Concerning our ‘...’ response to Reviewer #3’s suggestion to undertake another copy edit, we would like to apologize, since it was not intentional. It was a typo deriving from revision rounds among co-authors. We fully agreed with Reviewer #3 and the other two Reviewers about the necessity to undertake an improvement in writing style, and indeed we followed the Reviewers suggestions on this aspect.

My comments below are meant to strongly urge the authors to consider their use of the term "evolvability" and how they describe their views on convergence, and to make sure their text aligns with their own thoughts. That is, I think I may disagree, but I don't have any specific edits to suggest, and I don't think it is too terrible for the authors to opine as they wish.	We believe that by explicitly mentioning Jablonski in text (not only with the reference) we make clear that we are referring to the ‘phenotypic’ concept of evolvability (which is clearly different from evolvability in genetics, for instance). In the work that we cite, Jablonski treated this definition of evolvability in more detail and readers may want to read more on these aspects on the related publication. Moreover, we believe that by already introducing our idea of phenotypic convergence in the Introduction (as suggested by Reviewer #2) we emphasize our understanding of convergence so that the text aligns more with it.
I appreciate that the authors better justified their shape vs structure argument. I think they may have gone a bit rough on shape analyses, though. Certainly there is much we have learned about shape correspondence with functional regime. And, there are certainly cases of muscles, ligaments, etc. massively changing (e.g., bats; Bahlman et al. 2017 J Anat).	Our intention was not to underestimate the widely recognized correspondence between some shape traits and ecological adaptations but, instead, to stress that, if one has to compare how shape vs. structure respond to ecology during evolution, we expect the latter to respond more. To avoid ambiguities, we added two sentences in the Introduction (LL. 70-72) where we clearly state that for both the studied levels (hence, including shape) there is abundant evidence of ecologically driven features, and we also provide two textbook examples. As a consequence of this new addition, we changed one word in LL. 91: ‘Assessed’ (previous MS version) was changed to ‘Compared’, because in this new MS version, we mention some studies on the ecological signal borne by shape OR structure (hence studying one of the two levels) (those discussed above and cited in LL. 70-72) but in L.91 we want, instead, to talk about the rare studied that ‘compared’ the ecological signal borne by shape AND structure (hence studying both the levels). As for the study on bat’s musculoskeletal massive re-organization that Reviewer #3 kindly suggests, we now mention it in another Introduction section (LL. 76-77). Here we clarify that we expect large modifications in external shape (and the consequent reorganization of surrounding tissues) to be infrequent, although not impossible (and here providing the example suggested by Reviewer #3)
It also seems odd that there is no mention of the plasticity (i.e., within an individual's lifetime) that is inherent in the argument the authors are making (and is the focus of the Kivell into they suggested). Plasticity may be linked to evolvability, or it may inhibit evolvability--in both cases you might expect plastic traits to align with function in the way	We followed Reviewer #3’s suggestion adding an Introduction paragraph (LL. 81-88), treating potential relationships between ontogenetic plasticity and evolvability.

you tested in this study	
My other issue is with the added section in response to Grossnickle et al. If the authors wish to use a large portion of the Discussion to critique the Grossnickle et al. findings (which is fine), it would be helpful if they included a demonstrative figure illustrating what is on lines 337-346. As written, it is a bit ambiguous. For instance, what exactly does this mean: "if the distance between the corresponding reconstructed ancestral trait values happen to be longer than the observed values of focal lineages."? Do you mean a comparison between the distance(s) between, say, ancestral state values at nodes and the distances between tips? Or the values at the tips?	Following Reviewer #3's suggestion we included a figure (Supplementary Figure 8, extracting and modifying Fig. 4) that visualizes the two concepts of convergence. As described in the main text, we show how the distance between reconstructed ancestral values (D_{anc} in Supp. Fig. 8) is key to determine if the evolutionary trajectories converge or not (keeping the observed phenotypes in extant taxa constant, of course). The example that we present (detailed in Supp. Fig. 8) is a particularly simple case, because the directions to measure D_{anc} and D_{tips} are almost parallel (we added this clarification in the text and we rephrased, L. 365-369, new MS version). In this simplified case (shown in Supp. Fig. 8):  - if $D_{anc} > D_{tips}$: the trajectories are convergent; - if $D_{anc} < D_{tips}$: the trajectories are divergent - if $D_{anc} = D_{tips}$: the trajectories are parallel In more complex cases (e.g. if ab in Fig. 8 is far from being parallel to AB) $D_{anc} > D_{tips}$ may not be simply the condition to have convergent trajectories, but the main point still remains: i.e. D_{anc} value alone crucially determines if trajectories are convergent or not and, hence, under a strictly geometry-based definition of the process, if the two taxa evolutionarily converge or not. We believe that since ancestral states are reconstructed assuming Brownian Motion and ancestral states determine D_{anc} (that, in turn, determines if two taxa geometrically converge or not, as shown above), our approach (C indices + observations of focal vs. non focal taxa on phylomorphospace) is more suitable to this study.
I also don't quite understand -- isn't it necessary, for taxa to have "converged", for them to have ancestors that were more "divergent"? After all, sister taxa that all cluster together would not be considered convergent. In my experience the term is only used--either technically or colloquially--when considering taxa from different parts of a phylogeny, interspersed at the tips with taxa in different functional and morphological regimes (if considering functional-morphological convergence). I am not sure the authors mean to	As detailed in Supplementary Figure 8, our concept of convergence is not based on geometrical convergence. We think that this ambiguity may derive from a terminological issue. We believe that Supp. Fig. 8 and its caption clarifies our view and what Reviewer #3 rightly states in this comment: we are considering taxa from different parts of the phylogenetic tree that evolve similar morphologies.

suggest anything different, but this is how I read the paragraph, so they may want to consider clarifying.